



# Runoff sensitivity to spatial rainfall variability: A hydrological modeling study with dense rain gauge observations

Clara Hohmann[1,2], Gottfried Kirchengast[1,2,3], Sungmin O[2,3,4], Wolfgang Rieger[5], and Ulrich Foelsche[3,1,2]

[1]Wegener Center for Climate and Global Change (WEGC), University of Graz, Graz, Austria
[2]FWF-DK Climate Change, University of Graz, Graz, Austria
[3]Institute for Geophysics, Astrophysics, and Meteorology/ Institute of Physics, University of Graz, Graz, Austria
[4]now at: Biogeochemical Integration, Max Planck Institute for Biogeochemistry, Jena, Germany
[5]Bavarian Environment Agency, Germany

**Correspondence:** Clara Hohmann (clara.hohmann@uni-graz.at)

**Abstract.** Precipitation is a key input to hydrological models. While rain gauges provide the most direct precipitation measurements, their accuracy in capturing rain patterns highly depends on the spatial variability of rainfall events and the gauge network density. In this study, we employ a high-resolution meteorological station network (mean station distance of 1.4 km), the WegenerNet in southeastern Austria, to investigate the impact of station density and interpolation schemes on runoff simulations.

We first simulate runoff during heavy precipitation (three short-duration and three long-duration events) using a physically based hydrological model with precipitation input obtained from a full network of 158 stations. The same simulations are then repeated with precipitation inputs from subnetworks of 5, 8, 16, 32, and 64 stations, using three different interpolation schemes – Inverse Distance Weighting with a weighting power of 2 and of 3, respectively, and Thiessen polygon interpolation. We find that the performance of runoff simulations is greatly influenced by the spatial variability of precipitation input, especially for

short-duration rainfall events and in small catchments. For long-duration events, reliable runoff simulations in the study area can be obtained with a subnetwork of 16 or more well-distributed gauges (mean station distance of about 6 km). We find a clear effect of interpolation schemes on runoff modeling as well, but only for low-density gauge networks. The sensitivity to the precipitation input is smaller for long-duration heavy precipitation events and bigger catchments. As a next step we suggest to study an ensemble of precipitation datasets in combination with runoff modeling to be able to decompose the effects of

precipitation measurement uncertainties and its spatial variability.

## 1 Introduction

Heavy precipitation events can have significant impacts on society and ecosystems by causing severe floods and landslides. Moreover, they intensify under climate change in many areas (Kharin et al., 2007; Chen et al., 2012; Fischer and Knutti, 2016; Prein et al., 2016). Hydrological models have served as important tools to assess the impacts of heavy precipitation events

on runoff and hydrological processes. Since precipitation is the most important input in hydrological models (e.g., Bárdossy and Das, 2008; Zeng et al., 2018), it is crucial to understand its uncertainty and how the uncertainty in precipitation affects simulated runoff. Especially for heavy precipitation events, the spatial and temporal heterogeneity of precipitation becomes





more and more important. Convective storm cells with large volumes of precipitation can easily trigger hazards, but their limited spatial and temporal extent is associated with huge measurement uncertainties (Mcmillan et al., 2012). Beside the

measurement uncertainties, considerable uncertainty can arise when point-level measurements are spatially interpolated for final gridded products (Goodrich et al., 1995; Mcmillan et al., 2012; Huang et al., 2019; O and Foelsche, 2019). Such gridded datasets are crucial to obtain areal precipitation information within catchment and subcatchment areas, especially for spatially distributed hydrological models. Areal precipitation data can also be derived from radar and satellite-based observations. But all measurements come with their own uncertainties and pros and cons (Kidd and Huffman, 2011). On one side, the point

measurements are most reliable for the quantitative amount of precipitation sums. However, they often do not provide reliable spatial patterns of heavy precipitation, because of their sparse distribution. On the other side, radar systems show a high spatial resolution of the precipitation cells, but do not give specific precipitation amounts (e.g., Sun et al., 2000; Tetzlaff and Uhlenbrook, 2005). Satellites indirectly estimate precipitation and therefore their data are subject to errors and uncertainties (Tian and Peters-Lidard, 2010; Kirstetter et al., 2012; O et al., 2017; Lasser et al., 2019).

Despite the availability of remote-sensing data, ground-based precipitation measurements are still widely used in hydrological modeling (Lopez et al., 2015; Zeng et al., 2018). Many studies, like those by Lopez et al. (2015), Goovaerts (2000) and Zeng et al. (2018), also pointed out the advantage of dense and well-distributed precipitation station networks. Since many years the effect of different precipitation station densities in hydrological models has been analyzed (e.g., Obled et al., 1994; Dong et al., 2005; Bárdossy and Das, 2008; Meselhe et al., 2009; Xu et al., 2013; Zeng et al., 2018; Huang et al., 2019). Dong

et al. (2005) and Xu et al. (2013) used a statistical approach to identify the appropriate number of precipitation gauges and then focused on the influence on the model performance of a lumped model. Both studies found a threshold after which an increase in station density does no lead to better model performance. Such a threshold can also be seen in many other studies (e.g., Bárdossy and Das, 2008; Zeng et al., 2018). Meselhe et al. (2009) used a conceptual and a physically based model to identify the impact of temporal and spatial sampling of precipitation (highly dense station network) on runoff predictions. The physically

based model was more sensitive to spatial and temporal resolution of rainfall. A threshold with no significant increase of model performance can also be seen in this case for both models. Huang et al. (2019) used a lumped and a distributed hydrological model to study the sensitivity of model performance to spatial rainfall resolution. They found the most important aspect in the temporal resolution, with better model performance under higher temporal resolution. Many of these studies put the model performance in focus and used a lumped hydrological model. To keep the model uncertainty low, we used a process-based

model and tested the sensitivity of different precipitation station densities.

By using station data for hydrological models, also the spatial interpolation schemes come into account. Many different interpolation options and possibilities are broadly studied (e.g., Zhang and Srinivasan, 2009; Ly et al., 2013; Szcześniak and Piniewski, 2015). Widely used precipitation interpolation schemes are Arithmetic Mean (Andréassian et al., 2001; Dong et al., 2005; Huang et al., 2019), Thiessen Polygons (TP) (Kobold and Brilly, 2006; Meselhe et al., 2009; Zeng et al., 2018), Inverse

Distance Weighted (IDW) (Verworn and Haberlandt, 2011; Schwarzak et al., 2015; O and Foelsche, 2019; Breinl et al., 2020), and different types of kriging like Ordinary Kriging (St-Hilaire et al., 2003; Verworn and Haberlandt, 2011; Breinl et al., 2020) or External Drift Kriging (Bárdossy and Das, 2008; Verworn and Haberlandt, 2011; Xu et al., 2013). The differences





between the interpolation schemes are especially pronounced for extreme values (Szcześniak and Piniewski, 2015). Therefore, the selection of interpolation schemes can affect hydrological simulations especially under heavy and extreme rainfall events.

Because our study area has a moderate topography with well-distributed stations, we do not expect an added value from more complex geostatistical methods like kriging and therefore we decide to focus on the two most widely used interpolation schemes TP and IDW.

To overcome the uncertainties with the spatial and temporal resolution a highly dense station network of precipitation point measurements can help. But, how many stations do we need to reliably model hydrological runoff under heavy rainfall events?

And how large is the influence of interpolation scheme on runoff results under different station network densities? The highly dense station network WegenerNet (WEGN) in the southeastern Alpine forelands of Styria, Austria allows us to study these questions related to the Raab catchment. The region of southeastern Styria, Austria is well known for heavy precipitation events (Schroeer and Kirchengast, 2018; Breinl et al., 2020) (Sect. 2.1). Because of the data availability (Sect. 2.2) it is possible to analyze the influence of precipitation station densities on runoff in detail. Therefore, we set up the widely physically based

"Water Flow and Balance Simulation Model" WaSiM (Schulla, 1997) (Sect. 3.1) and simulated runoff under several setups (Sect. 3.2). We used different station densities/numbers from 5 to 158 stations, as well as different precipitation interpolation schemes with one TP scheme and two IDW schemes with a weighting power of 2 (IDW2) and of 3 (IDW3). Previous studies have assessed the impact of the station density and interpolation on precipitation data quality such as mean and extreme rainfall values (Gervais et al., 2014; Avila et al., 2015; Herrera et al., 2018). We focus on the impact of such precipitation

uncertainty on hydrologic simulations, especially runoff peaks and the combination of station density and interpolation method. We analyze three short- and three long-duration heavy precipitation events in summer (May to September) on the catchment (500-1000 $km^2$) and subcatchment (10 - 50 $km^2$) scale. The results are on the one hand divided in a section of individual example events with precipitation maps and runoff curves (Sect. 4.1). On the other hand, in ensembles of events where all catchments, events, and interpolation schemes are combined and the peak flow deviations are analyzed (Sect. 4.2). It is followed

by a detailed and combined discussion of the results (Sect. 5), and ends with conclusions and an outlook on further studies (Sect. 6).

## 2 Study area and data

### 2.1 Study area

The study area is part of the Raab catchment, a southeastern Alpine foreland river. The river Raab flows from the "Passailer"

Alps in the state of Styria, Austria at a height of around 1150 m a.s.l. to the Danube river in Hungary. The focus area of this study ranges from the gauging station Takern II/Raab to Neumarkt/Raab with a total area of around 500 $km^2$ (Fig. 1). The gauging station Feldbach/Raab is located in between. Beside the main river Raab also the tributaries with subcatchments of around 10 to 50 $km^2$ are of interest. Therefore, we choose to analyze five subcatchments (Tabel 1, Fig. 1). The subcatchments are all covered by the WEGN itself. Three subcatchments are on the northern side of the Raab and two are on the southern.

Since we do not have measured runoff data for these subcatchments, we implemented pour points in the model directly before





they flow into the Raab river. The Haselbach (12 km$^2$) is taken as a representative small subcatchment and the Grazbach (54 km$^2$) as a bigger one for our analysis. Both can be seen as typical subcatchments in our study area.

The total study area is moderately hilly with elevations from 230 m to 530 m and located in the southern alpine foreland. The land use is dominated by agriculture areas and patchy forests. The dominant soil type is sandy loam. The mean annual
precipitation is around 850 mm and the mean annual temperature about 9.5°C. The study area was chosen because of its vulnerability to heavy/convective precipitation events (Schroeer and Kirchengast, 2018) and climate change (Hohmann et al., 2018). The region is well equipped with a highly dense climate network, the WEGN, which was built up by the Wegener Center for Climate and Global Change, University of Graz, Austria (Kirchengast et al., 2014). The WEGN measures precipitation, temperature, humidity, and other variables since the beginning of 2007 with 150 stations (about one per 2 km$^2$, 5 min sampling)
in an area of 22 km x 16 km. All data are quality controlled by the WEGN QC system (Kirchengast et al., 2014) and additional bias correction is implemented for precipitation data, using the approach by O et al. (2018).

**Table 1.** Characteristics of the study catchment and representative subcatchments with the total basin area up to the gauging station/pour point in the river Raab.

| (Sub)catchment | Area [km$^2$] | Location to river Raab |
|---|---|---|
| Neumarkt/Raab (total catchment) | 987 | - |
| Neumarkt/Raab (focus area) | 488 | - |
| Feldbach/Raab (total catchment) | 689 | - |
| Feldbach/Raab (focus area) | 190 | - |
| Grazbach | 53.9 | north |
| Auersbach | 28.9 | north |
| Saazerbach | 27.2 | south |
| Haselbach | 12.3 | south |
| Kornbach | 12.2 | north |

## 2.2 Data

For hydrological modeling of the Raab catchment with WaSiM we need meteorological data for precipitation, temperature, relative humidity, wind speed, global radiation and air pressure aggregated at a 30 minutes time resolution. Table 2 provides
an overview about the maximum available station amount for each parameter, as well as its source. The WEGN with its dense station network and 5 minutes time resolution is build up in a rectangular grid, because of the comparability to climate models. It is in the middle of the focus area around gauging station Feldbach/Raab, but does not cover the total catchment. Therefore, we also need to include data from the Austrian Weather Service (ZAMG) with 15 minutes time resolution and





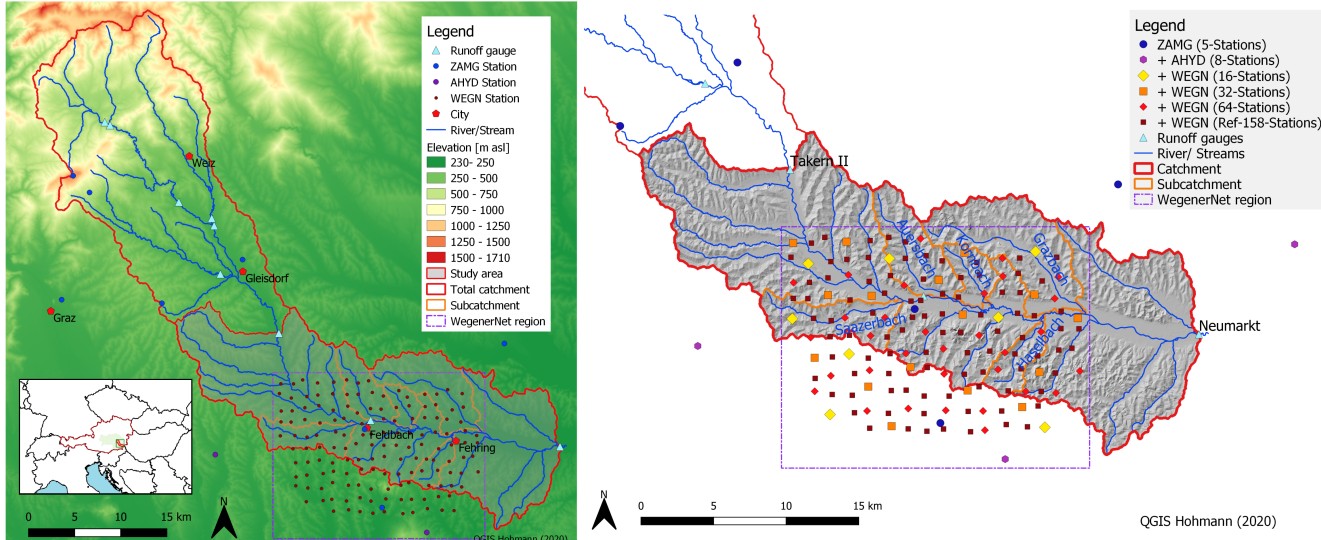

**Figure 1.** Map of the Raab catchment in southeastern Austria (left), including the full catchment down to runoff gauge Neumarkt/Raab (red line), the focus area (grey area), the subcatchments (orange line), the WegenerNet region (violet box), the locations of rain gauge stations (dot symbols) and the runoff gauges (triangle symbols). An enlarged view of the study area (right) indicates also more closely the used station subnetworks and subcatchments.

the Austrian Hydrographic Service (AHYD) with 1 to 15 minutes time resolution to properly simulate runoff (Table 2). To
run the model only for the focus area, gauging station Takern II/Raab is used as an inflow. Runoff data from gauging station Neumarkt/Raab are used for calibration, and Feldbach/Raab for cross checks and further analysis. For precipitation event identification the Integrated Nowcasting through Comprehensive Analysis (INCA) by Haiden et al. (2011), a multivariable analysis and nowcasting system developed at the ZAMG, is used.

Beside hydrometeorological station data, we need gridded data sets. The digital elevation model (DEM), river network, and
geological information are provided by the state government offices of the States of Styria, Austria (LStmk) and Burgenland, Austria (LBgld). The topographic analysis tool (TANALYS) of WaSiM uses the DEM to calculate other needed grids like flowtime, subcatchments, slope, river width and depth, etc. Homogeneous soil and land use grids (HYDROBOD) are provide by Klebinder et al. (2017) with a resolution of 100 m x 100 m for our research area. The HYDROBOD maps were created with the methods from Krammer et al. (2016). Maps for every single soil layer (0-20 cm, 20-50 cm, 50-100 cm) and parameter like soil
texture (percentage of sand, silt and clay), saturated hydraulic conductivity, Mualem van Genuchten parameters (combinations of residual water content and saturation water content), and soil thickness are used.





**Table 2.** Catchment attributes and hydrometeorological data used for the hydrological modeling with WaSiM with the following sources: HYDROBOD - homogeneous soil and land use grids by Klebinder et al. (2017), LStmk/LBgld – state government offices of the States of Styria/Burgenland, TANALYS – preprocessing tool of the hydrological model WaSiM, WEGN – highly dense station network data version 7.1 (Fuchsberger et al., 2019), ZAMG – data from the Austrian Weather Service, and AHYD – data from the Austrian Hydrographic Service.

| Catchment attributes | Source | Resolution |
|---|---|---|
| Land use types | HYDROBOD | 100 m |
| Soil information | HYDROBOD | 100 m |
| DEM | LStmk, LBgld | 10 m |
| River network | LStmk, LBgld | - |
| Geological information | LStmk, LBgld | - |
| Subcatchments, slope, river width & depth, ect | TANALYS output | 100 m |

| Meteorological data | Source | Number of Stations |
|---|---|---|
| Precipitation | WEGN | 150 |
| | ZAMG | 5 |
| | AHYD | 3 |
| Temperature | WEGN | 150 |
| | ZAMG | 5 |
| | AHYD | 3 |
| Relative humidity | WEGN | 150 |
| | ZAMG | 5 |
| | AHYD | 3 |
| Wind speed | WEGN | 12 |
| | ZAMG | 5 |
| Air pressure | WEGN | 1 |
| | ZAMG | 5 |
| Global radiation | ZAMG | 5 |
| Runoff | AHYD | 3 |

## 3   Modeling approach

### 3.1   Model setup and calibration

We used the hydrological model WaSiM, developed by et al., at the ETH Zurich in Switzerland for climate change studies
in Alpine catchments. WaSiM is a well-established widely used distributed and process-oriented hydrological model. It has





been used in similar catchments and for many different purposes like climate change studies (e.g., Bürger et al., 2011; Gädeke et al., 2014; Hohmann et al., 2018), land use changes (e.g., Alaoui et al., 2014; Yira et al., 2016) up to operational use (e.g. at BAFU Switzerland). We focused on a process-oriented model to keep the model uncertainty small, compared to lumped models, which are often used for similar precipitation runoff studies (e.g., Dong et al., 2005; Zeng et al., 2018; Huang et al., 2019). Furthermore, WaSiM was already successfully applied by Hohmann et al. (2018) in the study area for a climate change sensitivity study with a low flow focus.

In this study, we used the WaSiM Version: Richards-10.02.03. All used modules of WaSiM are shown in Fig. 2. For more information about the modules see Schulla (1997) or the WaSiM user guide by Schulla (2019). The model is set up with a spatial resolution of 100 m x 100 m and a temporal resolution of 30 min. WaSiM internally interpolates the meteorological station data to grids. The evapotranspiration is calculated after Penman-Monteith (Monteith, 1965) and the unsaturated zone with the Richards approach parameterized on the basis of van Genuchten (1980). For WaSiM the soil is split up in four calculation layers (0-20 cm, 20-50 cm, 50-100 cm, 1-20 m) with a total depth of 20 m, including the first groundwater layer. With the data from Klebinder et al. (2017), we end up with 416 soil parameter combinations in the soil table of WaSiM for our study domain.

The final groundwater parameters of the 2D groundwater module were fitted to represent the baseflow quite well during calibration period. Therefore, the saturated horizontal conductivity is split up in areas around the river with $5 \cdot 10^{-5}$ m s$^{-1}$ and surrounding hilly areas with $1 \cdot 10^{-6}$ m s$^{-1}$. The colmation factor is set to $1 \cdot 10^{-5}$ and the storage coefficient to 0.2 m$^3$ m$^{-3}$.

Beside the gridded groundwater parameters, WaSiM is calibrated with four parameters of the soil module, which influence shape and volume of the simulated runoff hydrograph and no measured or literature data are available (Schulla, 2019): The storage coefficient of surface runoff *kd* (shape of surface runoff hydrograph) and interflow *ki* (shape of interflow hydrograph), the drainage density for interflow *dr* and a recession constant of the soil *krec* in the soil table (both influencing the amount of interflow).

The model calibration period was from the 01.05.2009 to 30.09.2009 with a model spin up from 1.11.2007 to 30.04.2009. We calibrated the model only for the summer months (May to September), because summer months are in focus and the snow runoff is not of interest in this study for convective rainfall events in south-eastern Styria. The validation period was the summer of 2010 (01.05.2010 to 30.09.2010). The model performance was assessed with 50 % Nash-Sutcliffe efficiency (NSE) (Nash and Sutcliffe, 1970) and 50 % Kling-Gupta efficiency (KGE) (Gupta et al., 2009). The calibration was performed first with the shuffled complex evolution optimization algorithm developed at the University of Arizona (SCE-UA) (Duan et al., 1994) to get a first best guess of the model parameters. Second, to also include the physics behind the parameters and especially to include the distribution of the runoff components, the model was manually recalibrated: checking the NSE and KGE values, visually comparing the measured runoff with the simulated runoff and visualizing the runoff components for specific events. Because of the necessity of such manual recalibration, the model was calibrated with the IDW2 interpolation and 158 precipitation stations and not recalibrated with all different precipitation inputs and interpolation schemes. This setup is assumed to capture the spatial variation of precipitation in our study area.

The best model performance was obtained with the parameter set of *krec* 0.8, *dr* 9, *kd* 1.5 and *ki* 2. These setup results in a model performance for the river runoff in the calibration period of summer 2009 with an NSE of 0.79 and KGE of 0.76. The





validation period of summer 2010 results in an NSE of 0.67 and KGE of 0.81. After Moriasi et al. (2007) our model setup has a very good performance in the calibration period and a good performance in the validation period.

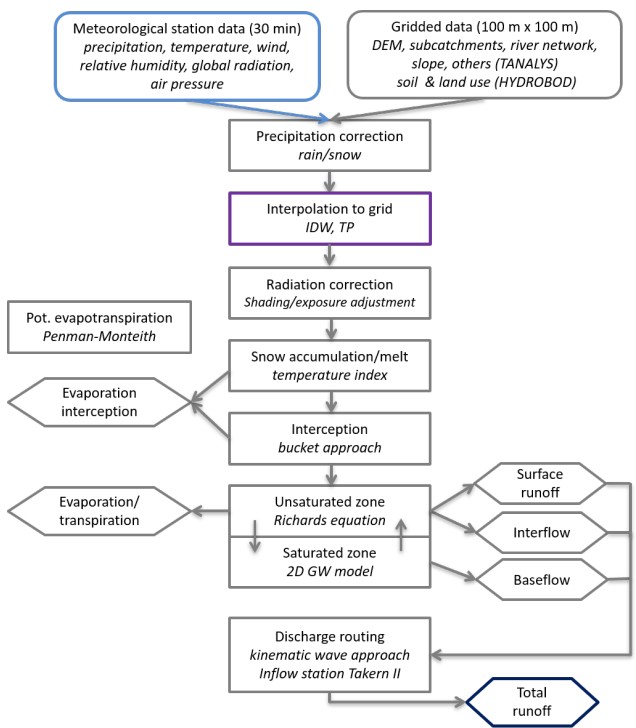

**Figure 2.** WaSiM model setup including modules and input data sets used in this study (created after scheme of Schulla (2019)). Focus of this study is to evaluate runoff output data (box marked in dark blue) simulated by WaSiM with various precipitation data resolutions at input (blue box) and using different interpolation schemes (violet box).

## 3.2 Experimental design

Our study design is visualized in Fig. 3. We are analyzing simulated runoff in different catchments and subcatchments (Sect.
2.1). Especially different station network densities for the precipitation input are in focus (Sect. 3.2.1). The six heavy precipitation events with focus on three short-duration and three long-duration events are analyzed in Sect. 3.2.2. The three different interpolation methods, which are already implemented in WaSiM, are explained in Sect. 3.2.3. The analysis with focus on runoff time series and peak flow deviation is presented in Sect. 3.2.4.

### 3.2.1 Selection of precipitation station network densities

To obtain precipitation input data at various spatial resolutions, we define six precipitation subnetworks consisting of different numbers of rain gauges ranging from 5 to 158 (Table 3). For instance, the lowest-density network (*5-Stations*) is defined using ZAMG stations only, with a mean station distance of 11 km. This could be a normal setup for operational use of hydrological





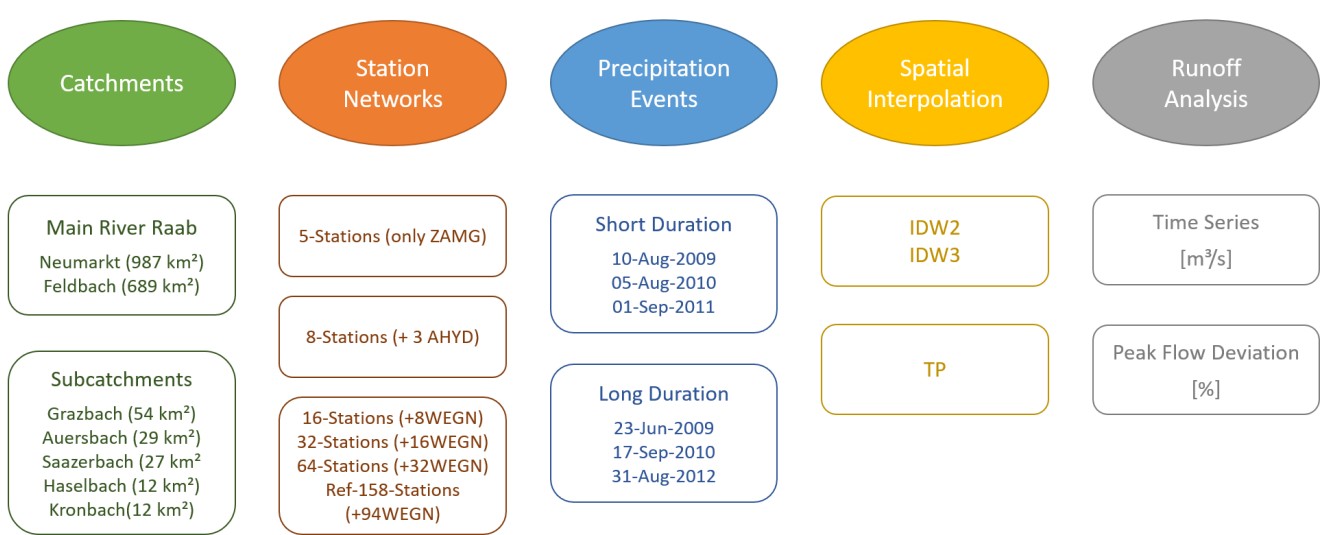

**Figure 3.** Overview of the study design with total catchment and subcatchments, gauge station subnetworks and full precipitation network, analyzed short- and long-duration events, spatial interpolation schemes (Inverse Distance Weighting with power of 2 (IDW2) and 3 (IDW3); Thiessen Polygons (TP)) for precipitation input data, and the key runoff output data analyzed.

models in Austria. When we include precipitation data from AHYD, with a plus of three stations, we get the *8-Stations* case and a mean station distance of 10.5 km. This would also be a typical setup for operational use, if a second source collects some

additional precipitation data. Then we double the amount of stations including eight WEGN stations (*16-Stations*), which are selected to provide a good spatial coverage of precipitation data and end up with a mean station distance of 6.2 km. We double the number of gauges to define the *32-Stations* and the *64-Stations* case, with a mean station distance of 4.0 km and 2.4 km, respectively. All available precipitation stations, 158 in total, are our reference (*Ref-158-Stations*) with a mean station distance of 1.4 km. We assume that the most accurate areal precipitation information can be obtained from Ref-158-Stations

and therefore we calibrated the model with this setup.

### 3.2.2 Selection of precipitation events

We selected heavy precipitation events among the top 10 % heaviest rainfall days during summer (May to September) within the 10-years period of 2007 to 2016 (O and Foelsche, 2019). Three small-scale short-duration and three large-scale long-duration events are selected through visual inspection of the WEGN and INCA data over the study area (Fig. 4). In Table 4 you find the

three heaviest short-duration precipitation events, as well as the three heaviest long-duration events. 2009 was the year with the heaviest events in our study period. The heaviest short-duration event (short-1) was on 10-Aug-2009 with 34 mm precipitation and a peak runoff at station Neumarkt/Raab of 107 $m^3 s^{-1}$, a $HQ_1$ event. The biggest event, the long-1 event measured from 22-Jun-2009 until 24-Jun-2009 with 121 mm precipitation lead to a peak runoff of 244 $m^3 s^{-1}$ at Neumarkt/Raab. This "long-1





**Table 3.** Precipitation station subnetwork cases with the total number of stations per subnetwork, together with the specific station data source (Z - ZAMG, A - AHYD, W - WEGN) and estimated mean station distance, the latter calculated with an ArcGIS tool.

| Gauge Subnetwork Case | Number (source) of Stations (Z/A/W) | Mean Station Distance [km] |
| --- | --- | --- |
| 5-Stations | 5 (5/-/-) | 11.0 |
| 8-Stations | 8 (5/3/-) | 10.5 |
| 16-Stations | 16 (5/3/8) | 6.2 |
| 32-Stations | 32 (5/3/24) | 4.0 |
| 64-Stations | 64 (5/3/56) | 2.4 |
| Ref-158-Stations | 158 (5/3/150) | 1.4 |

event" (23-Jun-2009) resulted in a flood peak bigger than a $HQ_{10}$. The other selected heavy precipitation events lead to runoff

peaks smaller than a $HQ_1$.

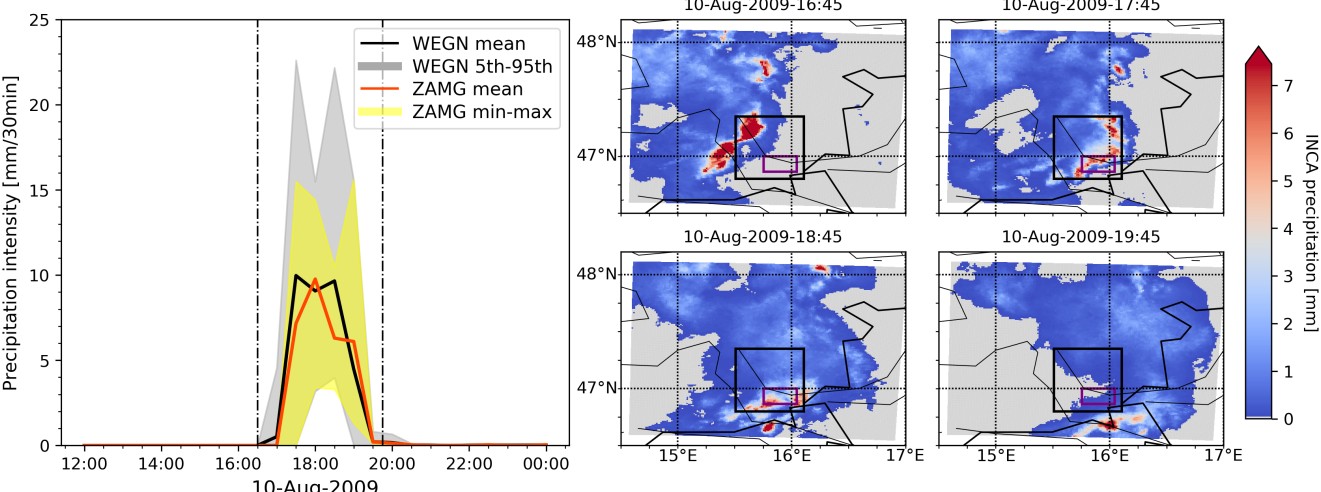

**Figure 4.** Precipitation time series of the "short-1 event" measured by WEGN and ZAMG stations (left panel) over the WEGN network area (red box in the four right panels). "WEGN" shows mean areal precipitation computed from the 150 stations (black line) with a 5th to 95th percentile range among the stations (gray shaded), while "ZAMG" shows mean precipitation (red line) obtained from the 3 ZAMG stations with a min-max range across the stations (yellow shaded). The maps sequence (four right panels) shows the evolution of the precipitation event as captured by the gridded INCA analysis over the WEGN network (red box) and the larger Raab catchment region (black box).





**Table 4.** Precipitation events selected for this study and associated key characteristics. The precipitation information indicated for the events is estimated from WEGN data. The runoff information is the measured peak runoff at gauging station Neumarkt/Raab.

| Event | Date | Duration [h] | Total precipitation [mm] | Peak hourly precipitation [mm] | Peak runoff [$m^3\,s^{-1}$] |
|---|---|---|---|---|---|
| short-1 | 10-Aug-2009 | 4 | 34 | 19 | 107 |
| short-2 | 05-Aug-2010 | 4 | 33 | 18 | 27 |
| short-3 | 01-Sep-2011 | 4 | 31 | 26 | 26 |
| long-1 | 23-Jun-2009 | 65 | 121 | 10 | 244 |
| long-2 | 17-Sep-2010 | 54 | 62 | 5 | 55 |
| long-3 | 31-Aug-2012 | 37 | 50 | 4 | 18 |

### 3.2.3 Spatial interpolation schemes

Several interpolation methods are implemented in WaSiM, e.g. TP, IDW, Elevation Dependent Regression, as well as different combinations (Schulla, 1997). In this paper, we test two different IDW setups and the TP, which are widely used interpolation methods in hydrological studies (Goovaerts, 2000; Ly et al., 2013; Szcześniak and Piniewski, 2015). We decided not to include height information for precipitation map creation, because the elevation differences in the area are fairly small (height differences no more than about 300 m). IDW is the sum of all contributing station data with specific weights (Schulla, 1997). It is calculated with the following equations (1) and (2):

$$\hat{z}(u) = \sum_{j} (w_j \cdot z(u_j)) \tag{1}$$

$$\text{with } w_j = \frac{1}{d(u,u_j)^p} \cdot \frac{1}{C} \text{ and } C = \sum_{j} \frac{1}{d(u,u_j)^p} \text{ follows } \sum_{j} w_j = 1 \tag{2}$$

$\hat{z}(u)$     interpolated value at location u

$wj$     weight of the observed value at the station j

$z(u_j)$     observed value at the station j

$d(u,u_j)$     distance to the station j

$p$     weighting power of the inverse distance

In our study we use the standard weighting power $p$ of 2 (IDW2) and for comparison also the weighting power $p$ of 3 (IDW3). In WaSiM, all stations in a specific radius are used for the interpolation. Only one specific search radius can be selected, which is then applied for all stations. In our study, we formally set the search radius to 50 km to be able to include





the surrounding weather stations also in subregions with larger station distances, which is necessary to get a robust coverage of
the total catchment area. With the TP interpolation scheme, always the precipitation data of nearest station are taken. So, each
grid cell of the model is getting the nearest station information and the formed polygons (Thiessen Polygons) are representing
lines of equal distance between two stations (Schulla, 1997). Hence TP is a simpler method than IDW, but still widely used in
hydrological modeling (Zeng et al., 2018; Meselhe et al., 2009; Kobold and Brilly, 2006).

### 3.2.4  Runoff analysis approach

In our study, we analyze the event-specific time series of runoff and peak flow deviation. Time series are visualized for all
events individually, but combined with different station network densities and interpolation schemes. For each catchment,
interpolation method and event, the peak flow deviation in percent is calculated individually. For this purpose, the maximum
runoff value is calculated for the simulation results of every subnetwork case (MAX value) and compared to the maximum
runoff value of the full-network reference case (MAX $\text{Ref}_{158Stations}$), which best captures the "true" spatial variability of
precipitation in the study area. This deviation metric is hence computed as follows:

$$\text{peak flow deviation [\%]} = \frac{(\text{MAX value}) - (\text{MAX Ref}_{158Stations})}{(\text{MAX Ref}_{158Stations})} \times 100. \tag{3}$$

## 4  Results

### 4.1  Results for individual example events

In this section we focus on individual precipitation events. Figure 5 shows an example map of the interpolated precipitation
data on the 100 m x 100 m grid of WaSiM, as well as the resulting runoff in the representative small subcatchment Haselbach
($12 \text{ km}^2$) for the short-1 event. The results of 5-Stations and Ref-158-Stations respectively with the interpolation schemes of
IDW2 and TP are visualized. With the 5-Stations case, the maps of the two interpolation schemes and the resulting runoffs
at Haselbach are very different. In the case of Ref-158-Stations the interpolation schemes have a smaller impact on the areal
precipitation estimation, compared to 5-Stations case. For the Haselbach at this short-1 event the difference between the IDW2
and TP interpolation under the 5-Stations case is more pronounced as the difference between the 5-Stations and Ref-158-
Stations cases.

In Fig. 6 the runoff time series of the short-1 event and the long-1 event for the interpolation schemes of IDW2 and TP are
visualized. The three columns show the results for all station densities for the small tributary Haselbach ($12 \text{ km}^2$), the biggest
tributary Grazbach ($54 \text{ km}^2$) and the total catchment Neumarkt/Raab ($987 \text{ km}^2$).

The **short-1 event** at Haselbach shows a special characteristic, because the setup with 8-Stations shows a second runoff
peak. Including also WEGN stations in the area (16-Stations case), the second peak is only marginally visible in the IDW2
case, but not under the TP scheme.





**Figure 5.** Precipitation maps using the WaSiM interpolation schemes of Inverse Distance Weighting with power of 2 (IDW2) and Thiessen Polygons (TP) for the short-1 event (10-Aug-2009 at 17:30), for the 5-Stations and Ref-158-Stations subnetwork cases (upper-left four panels). The time series (bottom row and right column panels) show the precipitation (dashed, from top) and the modeled runoff (solid) of this event in the representative small subcatchment Haselbach (12km$^2$).

Across the gauge density no systematic variation of simulated runoff peaks is observed. For instance, while the lowest runoff of 1.4 m$^3$ s$^{-1}$ is simulated from 8-Stations with IDW2, the highest runoff of 3.4 m$^3$ s$^{-1}$ results from 32-Stations with the same interpolation scheme. This is the same for the TP interpolation scheme. We find the lowest runoff of 1.6 m$^3$ s$^{-1}$ from the 8-Stations subnetwork, while the highest runoff of 4.1 m$^3$ s$^{-1}$ from the 32-Stations case. The spread between the lowest and highest runoff is around 75 % for IDW2 and 80 % for the TP interpolation scheme.

At Grazbach the IDW2 interpolation with 8-Stations also shows the lowest runoff with 6.4 m$^3$ s$^{-1}$. Less stations, so the 5-Stations case, result in more runoff with 7.4 m$^3$ s$^{-1}$. The other cases from the 16-Station case (9.3 m$^3$ s$^{-1}$) to the Ref-158-Stations case (10.9 m$^3$ s$^{-1}$) show increasing runoff with each step. Under the TP interpolation scheme the 5-Stations and





8-Stations cases are resulting in the same runoff with 6.6 m³ s⁻¹. The runoff with 32-Stations (9.9 m³ s⁻¹) is a little bit lower than the 16-Stations case (10.2 m³ s⁻¹). The cases with 64-Stations and Ref-158-Stations are almost the same. The spread between the lowest and highest runoff is for both interpolation schemes around 50 %.

For the short-1 event the Neumarkt/Raab catchment shows a very similar runoff curve for the two interpolation schemes of
IDW2 and TP. The Ref-158-Stations and 64-Stations cases show the highest runoff with 94 m³ s⁻¹ and 95 m³ s⁻¹ for IDW2 and 98 m³ s⁻¹ and 99 m³ s⁻¹ for the TP case. The lowest runoff with 56 m³ s⁻¹ (IDW2) and 54 m³ s⁻¹ (TP) results from the 8-Stations case. The spread between these runoffs is also for both interpolation schemes around 50 %.

The runoff of the **long-1 event** is more than twice as high as the one of the short-1 event. Also, the order and maxima and minima are very different between the two. The IDW2 and TP interpolation schemes lead to a different runoff curve order
and even different curve shapes. This becomes visible at Haselbach runoff curves with different shapes of the two interpolation schemes and different maxima and minima 6.0 m³ s⁻¹ (Ref-158-Stations) to 7.4 m³ s⁻¹ (5-Stations) for IDW2 and 5.1 m³ s⁻¹ (8-Stations) to 6.4 m³ s⁻¹ (16-Stations) for the TP interpolation scheme. However, the spread of around 20 % under both interpolation schemes, is very similar. Visually checked, especially the 5-Station case misses the first little peak under both interpolation schemes.

At Grazbach the lowest runoff under the IDW2 interpolation scheme is the 16-Stations case with 30 m³ s⁻¹ and the highest is the 5-Stations case with 34 m³ s⁻¹ , so a spread of 14 %. The TP interpolation scheme also has the lowest runoff with 31 m³ s⁻¹ under the 16-Stations case, and again the highest under the 5-Stations case with 39 m³ s⁻¹ , but it results in a spread of 25 %.

The spread at Neumarkt/Raab with 13 % (IDW2) and 20 % (TP) is quite similar to the spread at Grazbach, as well as the
highest values under the 5-Stations case (270 m³ s⁻¹ (IDW2) and 287 m³ s⁻¹ (TP)). But the order is different between the two. The lowest runoff is simulated under the 64-Stations case for IDW2 with 246 m³ s⁻¹ and the 158-Ref-Stations for the TP interpolation scheme with 237 m³ s⁻¹ .

These are examples of one short- and one long-duration event, for three catchments, but they do not cover the total range of setups and results. Therefore, combined figures are shown in the next Sect. 4.2.



**Figure 6.** Precipitation and associated runoff time series for the short-1 event (10-Aug-2009) (top six-panel plate) and long-1 event (23-Jun-2009) (bottom six-panel plate), respectively, for all five subnetwork cases and the full Ref-158-Stations network. Results are shown for the Inverse Distance Weighting with power of 2 (IDW2) and Thiessen Polygons (TP) interpolation schemes (top and bottom rows per plate), for the subcatchment Haselbach (left) and Grazbach (middle) as well as total catchment Neumarkt/Raab (right).





## 4.2 Combined results for all events

Figure 7 shows the peak flow deviations as calculated with Eq. (3) for all analyzed cases. It is visible that the different short- and long-duration rainfall events lead to a very different runoff picture, while the interpolation schemes and different catchments are much more similar. Lower station densities (e.g. 5-Stations, 8-Stations) mostly show a bigger deviation to the Ref-158-Stations full network case (darker colors) than cases with more stations (lighter colors). The three short events show more extreme differences (darker colors) between the station densities than the three long events (lighter colors). In the comparison between the (sub)catchments, no single (sub)catchment is especially noticeable.

The long-duration events show a slight tendency for overestimation of runoff (more blueish colors), while the short-duration events over- and underestimate (bluish and reddish colors). The TP interpolation scheme generally shows the most extreme values (darker colors) and the IDW2 the lowest variability. The runoff results for the IDW3 interpolation scheme lie in this respect between those from the TP and IDW2 interpolation schemes. The northern catchments Auersbach, Kornbach and Grazbach do not show differences if we simulate with 5- or 8-Stations subnetworks under the TP interpolation scheme, because of their location in relation to these station locations. Under the IDW2 and IDW3 interpolation schemes differences are present as expected.

For the **short-1 event** under all three interpolation schemes and for all (sub)catchments, the 8-Stations subnetwork shows the strongest negative peak flow deviation, amounting to near –45 %. The Haselbach subcatchment shows a slightly different behavior than the others, with a positive peak flow deviation for the 5-Stations case under the TP interpolation and the most positive value of around +30 % with the 32-Stations subnetwork. Compared to the short-1 event, the **short-2 event** shows a more positive peak flow deviation pattern. The 5-Stations and 8-Stations cases show a very different picture between the IDW and TP interpolation schemes, e.g., Saazerbach 8-Stations around –20 % with IDW2 and around –30 % with IDW3, but around +5 % with TP, or Kornbach 8-Stations IDW2 around 0 %, IDW3 +10 %, but TP +80 %. The **short-3 event** shows a very strong positive peak flow deviation for the northern subcatchments (Grazbach, Auersbach, Kornbach) and a strong negative peak flow deviation for the southern catchments (Haselbach, Saazerbach). This event also shows the strongest peak flow deviation of all events and all simulations ranging from -78 % to +220 %. The runoff gauges at the main river Neumarkt/Raab and Feldbach/Raab, seem to reflect the mix of extreme positive and extreme negative peak flow deviations with maximum values of around +30 % and –35 %.

The **long-1 event** shows very little peak flow deviations (smaller than around 15 %) with IDW2 and IDW3 interpolations, except for the Haselbach. Under the TP interpolation, the 5-Stations and 8-Stations subnetworks result in around +20 % to +30 % peak flow deviation for all the other catchments. The 16-Stations case at Kornbach shows a negative (around –10 % to –20 %) peak flow deviation under all interpolation schemes. The **long-2 event** looks very similar to the long-1 event, also with little peak flow deviations. The Auersbach shows the strongest peak flow deviations with values between around +25 % and –15 %. The 5-Stations and 8-Stations cases result in a positive peak flow deviation in all catchments. The **long-3 event** shows quite a mixed picture with positive and negative peak flow deviations. This event also shows the strongest deviations of the long-duration events. The Auersbach shows a negative peak flow deviation under all station subnetworks and interpolation





schemes. The Haselbach shows a peak flow deviation of +25 % for the 5-Station case under the IDW2 and IDW3 interpolation,
but no deviations for almost all other cases. Only the 64-Stations subnetwork under the IDW3 interpolation scheme result in a
–25 % peak flow deviation in these subcatchment.

By comparing **all events** together, some results can be summarized for individual (sub)catchments. For the river Raab
gauging stations Feldbach/Raab and Neumarkt/Raab, the peak flow deviation starting from the 32-Stations subnetwork is
almost the same as the Ref-158-Station full network (mostly smaller than 10 %). This implies that the 32 stations would be
enough for adequately enabling the simulation of the river Raab runoff. Under the long-duration events for the full-catchment
gauge Neumarkt/Raab, and using the IDW2 interpolation, even the 5-Stations or 8-Stations operational networks appear to be
sufficient, given that they exhibit less than around 10 % deviation. Overall, for the long-duration stratiform rainfall events, the
64-Stations subnetwork is almost everywhere as good as the Ref-158-Stations full network, independent of the interpolation
scheme.

In contrast, for the short-duration events and the comparatively small subcatchments, the station density is evidently much
more important, and the peak flow is much more event-dependent. For the short-1 event the 64-Stations subnetwork is almost
as good as the Ref-158-Stations full network (deviation smaller than 6 %). Only for the Haselbach, the 32-Stations subnetwork
under the TP interpolation scheme would not be enough (more than 30 % deviation); here, curiously, the 5-Stations case appears
better with only 13 % peak flow deviation. For the short-2 and short-3 events even the 64-Stations subnetwork would not be
as good as the Ref-158-Stations full network, given its almost 20 % deviation at Saazerbach (short-2 event) and Auersbach
(short-3 event).





**Figure 7.** Peak flow deviation to Ref-158-Station case as a grid-cell plot with each cell indicating the magnitude of the deviation on a color-scale, for the cases of all six events (figure panels), all (sub)catchments (columns per panel), all three interpolation schemes (stacked subpanels per panel) IDW2 and IDW3 - Inverse Distance Weighting with power of 2 and 3, TP - Thiessen Polygons, and all five subnetwork cases (rows per subpanel) analyzed in this study.





In Fig. 8 we visualize the summarized results for the peak flow deviations as a function of all station subnetworks. This summary view clearly highlights that the uncertainty in runoff simulations due to interpolation schemes and gauge network density is much greater for short-duration convective precipitation events. We further find that the direction of biases (overestimation 320 vs underestimation) is affected primarily by the gauge network density rather than the interpolation scheme. For long-duration heavy precipitation events, we find faster decreases in biases with increasing number of gauges in the network. 16 stations in our study area (around 500 km$^2$) yield satisfactory performance, with biases lower than 10 % for all subnetworks of at least this station number and all interpolation cases. Note that our subnetworks represent a quite regularly distributed gauge configuration, and therefore uncertainty in the runoff simulations can be somewhat greater for more irregular gauge location 325 configurations.

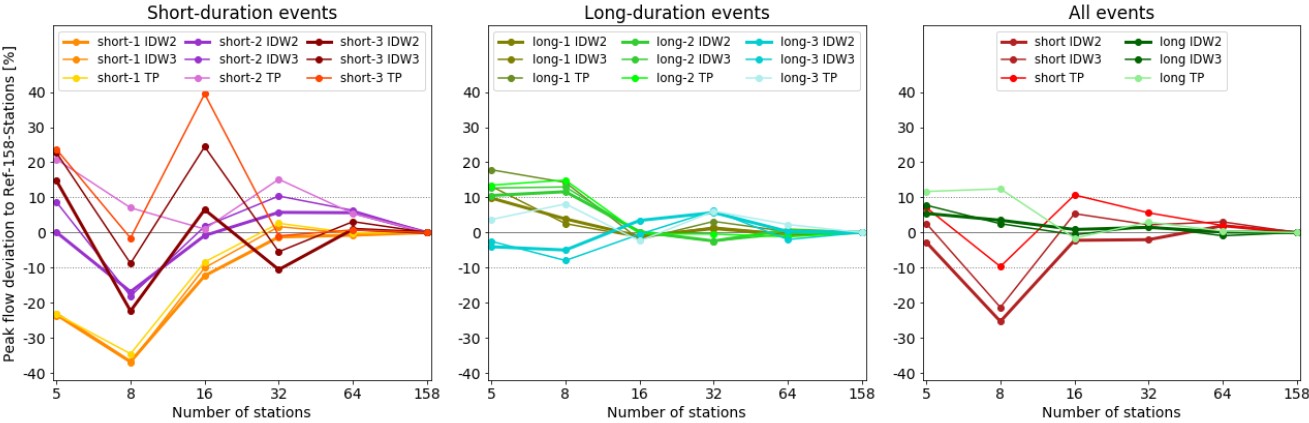

**Figure 8.** Peak flow deviation to Ref-158-Station case of the mean over all (sub)catchments for all subnetwork cases (x-axis of panels) and interpolation schemes (different line styles, see legend) IDW2 and IDW3 - Inverse Distance Weighting with power of 2 and 3, TP - Thiessen Polygons, respectively, for the three short events (left), the three long events (middle), and the mean each over the short and long events (right).

## 5 Discussion

Here we discuss the diversity of results of the station densities, interpolation schemes, (sub)catchments, and individual events in more detail and in synthesis. The mean over all catchments of the long-duration events shows a "sufficiency threshold" at the 16-Stations subnetwork, with just little runoff change ($< 6\%$) for more stations. This equals a mean station distance of 330 around 6 km, or around 16 stations per 1000 km$^2$. Beyond this station density, no strong further improvement of the simulated runoff can be observed in average over all catchments for long-duration events. In contrast, the mean over all catchments of short-duration events only show a "sufficiency threshold" at the 64-stations subnetwork, with just little runoff change ($< 6\%$) for more stations. Here a mean station distance of around 2.5 km, or rather around 64 stations per 1000 km$^2$ are needed.





Such thresholds go along with the literature, where no better performances after crossing specific station densities are seen
(e.g. Bárdossy and Das, 2008; Dong et al., 2005; Lopez et al., 2015; Xu et al., 2013). For example, Lopez et al. (2015) mention
an increase of performance with a denser station network up to 24 gauges per 1000 km$^2$, but no improvement after that for
up to 40 stations in the Thur basin (basin area around 1700 km$^2$). Xu et al. (2013) found, for a large-scale catchment, that the
performance leveled off after 93 stations (1 rain gauge per 1000 km$^2$), which was about 50 % of the 181 available stations in
the catchment of the Xiangjiang River (94 660 km$^2$). They also noted that below 38 stations (0.4 rain gauges per 1000 km$^2$)
the model performance was pretty poor. Dong et al. (2005) found in their catchment, Qingjiang river (12 209 km$^2$) a critical
number at 5 out of 24 precipitation stations.

In contrast, in our high-resolution case, the individual fairly small catchments do not show such an expected threshold,
especially for short-duration events. However, also the long-duration events do not show a salient threshold in all catchments
or events. Given its density of 1 station per 2 km$^2$, the station network in particular of the WEGN area is much denser than
any of the other networks studied. These studies used station densities such as about 2 station per 1000 km$^2$ Xu et al. (2013)
or up to 12 stations per 1000 km$^2$ Lopez et al. (2015). Our study hence detects and highlights the strong catchment and event-
dependence of the precipitation densities especially for short-duration events at 1-km-scale spatial resolutions. Therefore, for
proper modeling of the runoff from heavy convective precipitation events a highly dense station network is very important.
This was also seen to some degree by St-Hilaire et al. (2003), again a study that addressed larger scales, where areas with
high precipitation were better defined by denser networks for the long term (total annual precipitation) and short term (summer
convective events).

For our three short-duration events, the total precipitation amount of 31 mm to 34 mm is very similar, but it leads to different
simulated runoff curves. The short-1 event has a maximum peak flow of 107 m$^3$ s$^{-1}$, while the short-2 and short-3 events are
similar but smaller with 27 m$^3$ s$^{-1}$ and 26 m$^3$ s$^{-1}$. Overall, the precipitation amount might lead to different runoff curves,
depending on the location of gauges and storm core (O and Foelsche, 2019). From the runoff modeling point of view, it also
depends on the specific station locations and the measured precipitation amount at specific stations. This becomes clear with
the huge over- and underestimations of peak flow, depending on the different station densities (Fig. 7). The three long-duration
events show different runoff peak with 244/55/18 m$^3$ s$^{-1}$ and a total precipitation amount of 121/62/50 mm, respectively. The
peak flow deviations are very similar for the long-1 and long-2 event. The most stratiform event with smallest hourly peak
flows (long-3 event) shows a different picture, even though the total precipitation is similar to the long-2 event. In summary, we
can learn from this that in small catchments for short- and long-duration heavy precipitation events the amount of peak runoff
and of total precipitation are not directly related to the level of observed peak flow. The latter are driven by the specific event
characteristics in a more complex manner.

We emphasize that the explicit study of the hydrological response to different precipitation events is crucial. Many earlier
studies have evaluated the "accuracy" of (remote-sensing) gridded rainfall event data through direct comparison with ground
gauge measurements (e.g. O et al., 2017; Kirstetter et al., 2012; Lamptey, 2008). Now this study highlights that it is also im-
portant to evaluate the performance of precipitation datasets with various resolutions in terms of hydrological runoff response.





Such evaluation will provide practical guidance more widely both to rainfall data providers as well as to hydrological model users.

If the gauge network is sparse, positive bias values dominate the peak flow deviation for long-duration events. Sparse station data extrapolate too "heavy" into data-void areas. The short-duration events do not show this one-sided effect, both over-estimation and underestimation occur. The runoff for the short-duration events is much more event-depended, where some precipitation stations have a positive or negative influence. This reflects the more complex space-time structure of small-scale convective rainfall events. It goes along with the study by Lobligeois et al. (2014), where they analyzed that rainfall-runoff

processes are strongly variable between the catchments and precipitation events.

Over all events, (sub)catchments, and interpolation schemes very salient is the strong effect of the increase from the 5-Stations to the 8-Stations subnetwork, where on top of the ZAMG stations the AHYD stations are included. This sparse network change shows a particularly big variability range for small catchments, sometimes even with a second runoff peak (Haselbach). Especially the AHYD station Waltra, with its location closely south of the WEGN domain, has a strong influence.

In general, it was recently found, as well based on WEGN data, that the uncertainty in convective precipitation measurements is roughly exponentially decreased with an increase of gauge numbers (O and Foelsche, 2019).

As is expected for the small subcatchments, the station density has a bigger influence than for the total Raab catchment as observed at the runoff gauges Feldbach/Raab or Neumarkt/Raab. Also, the specific spatial location of the precipitation stations is much more important for small catchments. It has already been noted in other studies that the location of the precipitation

measurement is important on all scales (e.g. Lopez et al., 2015; Obled et al., 1994; Beven and Hornberger, 1982) but as we find here it again significantly increases for small catchments (10 $\text{km}^2$ to 50 $\text{km}^2$ area).

Turning specifically to characteristic influences of the interpolation scheme, several aspects are salient, including the special properties of TP interpolation. The subcatchments north of river Raab (Auersbach, Kornbach and Grazbach) are not affected by the AHYD stations (8-Stations case) under the TP interpolation because these stations are located west, south, and far east

of the study area. With the IDW2 and IDW3 interpolation scheme all stations within the search radius are included. Therefore, the AHYD stations are somewhat influencing the interpolated precipitation and thus also the modeled hydrological runoff. This well exemplifies the property that the precipitation maps of the gridded rainfall from the IDW and TP interpolation schemes are generally very different (Fig. 5). The root cause is the methodological difference that the borders between polygons of the TP interpolation are very stiff.

In extreme cases of high spatial rainfall variability, there are sharp differences of precipitation amounts between one polygon of TP interpolation and its neighbor. If one depends on few-station networks this "extreme" behavior of the gridded precipitation input becomes clear under TP interpolation, where the peak flow deviation is much more pronounced (Fig.s 7 and 8). The individual stations are much more influential compared to using the IDW interpolation schemes. Among the IDW schemes, the peak flow deviations of IDW3 are similar to IDW2, but with a tendency of IDW3 to be closer to TP interpolation. This occurs

because the higher IDW3 weighting power gives less weight to surrounding stations than IDW2, driving IDW3 results towards TP results where surrounding stations do not receive any weight.



Overall, if only a few stations are available, the IDW interpolation schemes, and in particular IDW2, are much more reliable. The TP interpolation scheme is not recommended for areas with complex topography and low station densities, in line with findings of Kobold and Brilly (2006). Here we note that the IDW interpolation schemes in our study always include all stations,

even the stations which are quite far away. This is adopted since WaSiM can only handle a single value as the IDW influence distance for all stations Schulla (2019). This value needs to be large enough to include relevant surrounding stations all over the study area (and was hence set to 50 km in our case). It would be beneficial, however, to change this value depending on varying neighborhood station distances across the network, which may further improve the gridded interpolation results from IDW.

Dirks et al. (1998) already recommended the simple TP interpolation scheme while using a spatially dense station network. Given the runoff simulations in our study does not show a strong interpolation scheme dependence under the Ref-158-Stations full network, it points as well into the direction that TP interpolation is sufficient for sufficiently dense networks. That a more advanced interpolation scheme does not necessarily show better performance under a high-density network was also found in the studies of Borga and Vizzaccaro (1997) and Syed et al. (2003).

In summary, the influence of the interpolation scheme is clearly visible, especially for few-station networks, but even there it is often less pronounced than the one of different station densities. Therefore, the impact of station network density is clearly much more significant for runoff simulations than the one of reasonably chosen interpolation schemes.

## 6 Conclusions

We used the highly dense station network WEGN in the southeastern Alpine foreland of Styria, Austria. In addition to that

eight stations of the Austrian operational station network, were used to analyze the influence of rain gauge network density and interpolation schemes on simulated stream and river runoff, with a focus on small catchments (10 km$^2$ to 50 km$^2$). We calibrated the hydrological model WaSiM with 158 precipitation stations (full network) and performed simulations based on short- and long-duration rainfall events. We use a cascade of subnetworks, ranging from 5 and 8 operational stations to 16, 32, and 64 station subnetworks, together with the widely used IDW and TP interpolation schemes. We find that our first

key question "How many stations do we need to reliably model hydrological runoff under heavy rainfall events?" cannot be answered in general for small catchments, due to the complex spatiotemporal characteristics especially of short-duration convective events. The influence of the station network density is specifically catchment- and event-dependent, but we were able to derive average guideline results.

For long-duration stratiform-type events (lasting typically longer than a day) and in average over all catchments a station

density with a mean station distance of around 6 km (16-Stations network in our area) is found to be sufficiently dense for robust runoff modeling including reliable peak runoff estimation. This station density is even significantly higher than the WMO recommendation for a minimum of one station per 250 km$^2$ in mountainous areas WMO (2008), corresponding to a 16 km mean station distance. For the average over all catchments from the short-duration heavy convective rainfall events (lasting typically a few hours only) we find at least a 64-Stations network is needed for runoff modeling with reliable peak





runoff estimation (mean station distance of around 2.5 km). Especially for short-duration convective events a dense station network (or other dense high-quality rainfall data product) is crucial.

The second key research question "How large is the influence of interpolation scheme on runoff results under different station network densities?" can be answered along with a strong station density dependency. For very dense station networks (in our case 64 to 158 stations, mean distance about 1.4 to 2.4 km) the specific interpolation scheme is relatively unimportant, as long as

reasonably chosen. Therefore, we find the simpler TP interpolation scheme already sufficient. In contrast, the sparse operational network sizes (in our case with 5 to 8 stations, mean distance about 11 km) perform better under the IDW2 interpolation scheme. Under such a sparse station number we find that including a few stations more (e.g., 5 or 8 stations) can result in very different runoff curves, even spurious secondary runoff peaks, in small catchments. Overall the interpolation scheme is found clearly less influential than the gauge network density on simulated runoff. Hence when analyzing and interpreting modeled

runoff based on rainfall input data most importantly the station network density influences the results, as long as a reasonable interpolation scheme is chosen.

In line with expectations, the larger Raab catchment, observed at runoff gauges Feldbach/Raab (689 km$^2$) and Neumarkt/Raab (987 km$^2$), appears to smooth over and compensate some of the extreme cases of smaller subcatchments. That is, the dependence on specific rainfall event characteristics and station network density is mitigated in the main river runoff. For many

local-scale hazards such as severe overland flooding, flashfloods, and hillslope landslides triggered by short-duration convective events, more dense observations are critical for reliable hydrological modeling for hazards risk estimation and protection. While the WEGN is a unique long-term research facility of sufficiently high station density, it is quite limited in area. Densification and expansion of runoff and rainfall gauge networks in this and many other risk-prone areas, would therefore be a great and much needed improvement on top of existing observations. An alternative are other data sources enabling suitable data

products at high spatiotemporal resolution such well calibrated high-quality precipitation radar data.

In deploying new stations, the selected station locations have a strong effect on gridded precipitation fields and therefore also on the runoff results, especially in small catchments. In this study we have selected subnetworks of gauges from the WegenerNet with quite regular distribution, given we had a quite flexible basis to choose from. A more detailed analysis, with random picking and evaluating more closely also irregular distributions, may be a useful planning and design step for new

station placements in other areas. This would help to arrive at an optimal rain gauge network design for hydrological purposes.

Since in almost no other places worldwide such dense networks are available, the runoff impact results arrived here for the Raab catchment and its subcatchments in southeastern Austria need to be "transferred" to other regions with due care of comparability of weather, hydrology, and landscape characteristics (cf. Kirchengast et al., 2014; Schroeer and Kirchengast, 2018; Schroeer et al., 2018). With such due care, we consider the essential results and conclusions transferable to many other

middle latitude land regions. For the huge number of ungauged or extremely sparsely observed small catchments, the awareness of both level of skill and limitations of rainfall-runoff modeling as reported here will be particularly crucial.



*Code and data availability.* The hydrological model WaSiM is available via http://wasim.ch (version used: Richards-10.02.03). The WegenerNet data are available at https://doi.org/10.25364/WEGC/WPS7.1:2019.1 (Fuchsberger et al. 2019: WegenerNet climate station network Level 2 data version 7.1 2007-2018; Wegener Center, Univ. of Graz, Austria). ZAMG data are available from the Austrian Weather Service
(www.zamg.ac.at) and AHYD data from the Austrian Hydrographic Service (https://ehyd.gv.at). HYDROBOD data are used from Klebinder et al. (2017) and are available on request from these authors. Geoinformation data are from state government offices of the States of Styria and Burgenland and are available from the respective GIS services (www.gis.steiermark.at; https://geodaten.bgld.gv.at).

*Author contributions.* All authors designed the study, with primary contributions by CH and GK. CH collected the data, performed the modeling and most of the analysis, created the figures, and wrote the first draft of the manuscript. GK provided guidance and advice on all
aspects of the study, and significantly contributed to the figure design and the text. SO contributed to the data collection, precipitation data analysis, and figure creation. WR supported the model calibration and validation. All authors helped to shape the research and analysis, they provided critical feedback and contributions to the text until submission and during review.

*Competing interests.* The authors declare that they have no conflict of interest

*Acknowledgements.* The authors thank J. Fuchsberger (Wegener Center, Univ. of Graz) for support and advice on the WegenerNet data, S.
Birk (Inst. for Earth Sciences, Univ. of Graz) for discussions and advice on the water balance and groundwater characteristics of the catchment, and V. Hess (DK Climate Change, Univ. of Graz) for several fruitful discussions and support during the study work. Furthermore, we acknowledge the data providers at the Austrian Weather Service (ZAMG), the Austrian Hydrographic Service (AHYD), the state government offices of Styria and Burgenland, and the Wegener Center regarding the WegenerNet data. WegenerNet funding is provided by the Austrian Ministry for Science and Research, the University of Graz, the state of Styria (which also included European Union regional development
funds), and the city of Graz; detailed information can be found online (www.wegcenter.at/wegenernet, last access: 29 August 2020).





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
