# Peer review of "Runoff sensitivity to spatial rainfall variability: A hydrological modeling study with dense rain gauge observations"

_Hydrology and Earth System Sciences, 2020_

## Referee Comment (RC1) · Anonymous Referee #1 · 20 Oct 2020

In their paper, Hohmann et al. studied the runoff sensitivity to spatial rainfall variability, namely they investigated the impact of station density and interpolation schemes on runoff simulations. I found the novelty of the study very limited. The effects of the density of the network of rain gauges and different interpolation methods on hydrological models have been analyzed many times in the past – as the authors mention themselves in the introduction to the paper. The research questions asked in the paper ("how many stations do we need to reliably model hydrological runoff under heavy rainfall events?" and "how large is the influence of interpolation scheme on runoff results under different station network densities?") are simply not new. I found the results to be very limited following the decision to explore only two interpolation methods (TP

and IDW), which seems to be motivated mainly by the fact that these are the interpolation methods that are built-into the WaSiM model that was used in this study while ignoring other common interpolation methods (e.g. Kriging). Are six rainfall events enough for this type of study? Considering the authors discuss the effects of rainfall variability in space, I would expect many more events to be interpolated and simulated to demonstrate the robustness of the results. I cannot recommend the manuscript for publication in HESS in its current form. I would suggest extensive revisions of the study that include extending the number of extreme rainfall events in the analysis, exploring additional interpolation methods and, as the authors suggested in the abstract, conducting further investigation to study if a rainfall ensemble can be used to decompose the effects of rainfall records and interpolation uncertainties on modelled runoff – this might be qualified as a novel aspect that can contribute to the originality of the study, which is now missing.

Some specific comments

Line 150. What is the reason to calibrate the model with an objective function that includes both NSE and KGE? Why not simply using one of the two?

Lines 156-157. The fact that the model is calibrated using IDW2 is not affecting the results when compared later to input using IDW3 and TP? I would expect the model to be calibrated to each input separately to reduce the errors emerging from the model parameterization (as your goal is to discuss the errors emerging from the model input).

Section 3.2.2. I suggest also comparing the rainfall events using some statistics. For example, how similar how the short rainfall events in term of their spatial variability? You can, for example, plot a graph comparing the spatial autocorrelation of the storms.

Section 3.2.4. Why using only a single metric to analyze the runoff? Consider adding other metrics, pointing on the timing of the peak runoff, total runoff volume, etc.

Lines 405-406. One option to overcome the model default is to compute the IDW

outside the WaSiM model and a second option can be changing the code of WaSiM to compute the IDW the way you choose (if I recall the model as an open code policy).

---

## Referee Comment (RC2) · Anonymous Referee #2 · 23 Oct 2020

This study tried to explore the effect of interpolation methods and station density on runoff simulation. However, this work has some severe problems and does not reach the standard of HESS in the current form. My comments are as follows.

Major comments:

1. It can hardly say that this study is novel or contributes much to our knowledge. The authors used two simple and widely used interpolation methods, TP and IDW. Results and conclusions are generally similar to previously published work. Besides, as a study focusing on the effect of station density and interpolation methods on hydrological modeling, the design is too simple. It is better and not difficult to (1) design station

networks with random numbers and locations of stations instead of using pre-defined networks (Table 3), and (2) conduct a sensitivity test of the weighting power instead of using 2 and 3. A negative example caused by (1) is that the authors state "32 stations would be enough ...", but this resulted from a large jump from 16 to 32 to 64 stations.

2. A large part of the study area is not covered by 158 rain gauges. Both IDW and TP cannot achieve reasonable estimates outside the WEGN network. As a result, only rainfall in the middle reach is largely affected by different density and interpolation methods, while rainfall in the upper and lower reach could always contain much larger errors. Meanwhile, the hydrological model is built over the whole domain, and runoff at catchment and sub-catchment outlets is compared. The effect of biased estimates in upper and lower reach on the runoff simulation could be very large.

3. Please perform the analysis based on a larger collection of precipitation events. Currently, only three small-scale short-duration and three large-scale long-duration events are selected. Although the six events may be representative, a comprehensive view in a long historical period is useful and necessary to demonstrate the all-aspect effect of interpolation and station density. The current results and analyses are all based on those limited events, making the results more "casual" than "causal". For example, the interpolation based on 5 stations could be largely affected by the location of storm centers, and thus the results based on 100 events could be different with results based on 6 events.

4. Figure 5: The IDW precipitation map based on 158 stations looks quite unrealistic. An actual precipitation event should be spatially continuous like that in Figure 4. However, the bull eye effect in Figure 5 is too obvious. The authors stated that all stations within a 50 km searching radius would be used in IDW. 50 km is quite large based on the measuring scale in Figure 1. This makes the bull eye effect even weirder.

5. The authors strengthened "spatial rainfall variability" in the title. But the spatial variability is not explicitly analyzed in this study. The title does not reflect what this

study actually did, i.e., station density and interpolation methods. Besides, the effect of station density and interpolation methods cannot be simply represented using "spatial variability".

Specific comments:

6. Please adjust the font of units which is often different from that of texts.

7. Line 25: It is better to state the "measurement uncertainties of rain gauges" because other approaches of rainfall measuring are not mentioned here.

8. Line 32-34: The pros and cons of radars and satellite sensors are similar in many cases and should be stated together.

9. Figure 1: It will be helpful to add latitude and longitude.

10. Line 124: Please complete the reference for WaSiM.

11. Line 150: Although I understand what you mean by 50% NSE and 50% KGE, please rephrase to be more formal.

12. Line 155-162: Given the authors state that manual recalibration is necessary, please add some descriptions on the benefits of manual recalibration. For example, what's the KGE and NSE before manual recalibration? Besides, the NSE decreased from calibration to validation periods but KGE increased. Please add some explanations.

13. Line 167: Strictly speaking, you used two, not three methods. Two different parameters do not make IDW two different methods.

14. Figure 4: Please use the shapefile of the catchment to replace the black box, which can help identify whether storm centers are located within or outside the river basin.

---

## Referee Comment (RC3) · Anonymous Referee #3 · 27 Oct 2020

**General comments and recommendation**

The manuscript by Hohmann et al. presents an analysis of the effect of different meteorological network densities, as well as different interpolation schemes on the simulated runoff for a meso-scale catchment in southeastern Austria. While indeed many questions about the optimal meteorological network density, about the adequacy and representativeness of stations' distribution and about the suitability of different interpolation schemes for hydrological modelling have not yet been fully answered, I do not see right now in this study an adequate and robust assessment, offering an added value to what we already know. The setup and design of the experiment is not thorough enough to support reliable statements with evidence.

The manuscript has per se a clear structure, the methods are generally described in a comprehensible way or supported by relevant sources, however, some explanations could be more clear and concise (e.g. the calibration procedure). Even though the discussion provides some good points, generally there are quite a few redundant paragraphs, while more interesting and critical points are not examined closely enough.The manuscript generally features high-quality and interesting figures; some of the tables and their captions should be reorganized in a more meaningful and efficient way. I would also suggest a native speaker to read it and correct it, some sentences definitely need to be rephrased.

Because of these considerations, I think the manuscript requires and extension of the experimental design, a more critical discussion and further work, before it can be possibly recommended for publication.

Please find my specific and technical comments here following.

**Specific comments**

- Experimental design:
  - One important drawback of your setup is the fact that despite the stations' density is high, it is not covering your whole study area, but only its central part.
  - You have "fix" a priori chosen subnetwork configurations, but for a fair evaluation of the effect of stations' subnetwork density you should try different (random?) configurations for the same number stations. Tentatively, you could also consider smaller jumps between one configuration and the next one.
  - You chose 2 interpolation methods (I wouldn't refer to three methods, as you simply changed a parameter of the second method), I think it would be more appropriate to use more interpolation methods, such as kriging, etc.. for a more robust evaluation of the effect given by the interpolation method, and also definitely to use a smaller searching radius (second spurious peaks in your hydrological simulations are not surprising, given how you interpolate precipitation).
  - The number and sample of events you are analyzing is simply too limited to allow you to make any meaningful assessment, now your analysis might show only very specific and localized effects, and might not hold for a larger ensemble of events. Is there any way to increase the number of events you analyze? You say you selected first heavy precipitation events among the top 10% heaviest rainfall days during summer, and out of these you selected through visual inspection your 6 events. I would suggest you rather select the top 100 events for 1d and 3d accumulation periods, for instance, and further analyze these?
  - (You are only using discharge gauging stations on the main river trunk, but you analyze also the contributing subcatchments. Sure, this is fine for looking at the effect of the different precipitation inputs, but not enough to disentangle the effects possibly stemming from the parameters of the hydrological model. You implicitly assume the hydrological model is working equally well on much smaller subcatchments, with the same parameters. As the

model is process oriented I am fine with this assumption, but you might want to spend a few words on this?)

- Data:
  - You have a 10 year period to choose from resp. analyze, is there some way to extend it?
  - You only report the return period for discharge, but actually it would be interesting – and relevant? - to know the return period associated with your rainfall events too.
- Model setup:
  - P7-L128-130: Lumped doesn't necessarily mean that a model is not process-oriented. I guess you mean conceptual?(that are often lumped, but not always)
- Calibration of the hydrological model
  - Please describe better and provide more details on your calibration procedure.
  - Why do you calibrate the model only basing on one summer, and only for one interpolation method?
  - Why do you use "only" NSE and KGE to define your objective function, instead of including further goodness-of-fit measures and criteria, perhaps more specifically tailored for floods, their volume and/or their timing?
- Runoff analysis approach:
  - I would recommend you expand your analysis approach by including also the peak timing, as this also is an important aspect in your modelling exercise, e.g. considering the effect of superposition of peaks simulated in the subcatchments.
- Results:
  - P16-L271: I wouldn't say that, e.g. Kornbach is rather systematically overestimated.
  - P16-L275-277: this is not true, see for example the short-2 event or the long-3 event.
- Discussion:
  - P 20: it is confusing you mentioning first Lopez et al.2015 report no increase in performance after including 24 gauges per 1000 km2, and later on saying they actually used 12 stations per 1000km2. They used gridded products to be able to increase the number of stations with "hypothetical" gauging stations. I think you should either specify this, or not go so much into detail.
  - P20-L360-363: Rephrase please this last paragraph.
  - P21-L380-381: Isn't this possibly case specific? Refer also to Lobligeois et al. 2014: In all regions, natural variability allows for contradictory examples to be found, showing that analyzing a large number of events over varied catchments is warranted.
  - I think you should expand more on the representativeness of stations' location.

**Technical corrections**

- P6-L124: developed..by who et al.?
- P7-L128: BAFU is FOEN in english
- Figure 2: shouldn't you remove the box with snow accumulation/melt, as far as I understood you are not using it?
- Figure 4: You should be consistent, and either use the min-max range in both cases, or the 5th and 95th percentile range. And why don't you also show the same kind of information for the 3rd source of precipitation data?
- Figure 6: Why don't you show observations in the last column?(i.e. for the Neumarkt/Raab gauging station)
- P17-L301 ..in these subcatchment should be in *this* subatchment
- P21-L373: ..more event-depended should be event-*dependent*

---

## Author Comment (AC1) · 13 Nov 2020

**Response to Referee #1**

*We thank the reviewer for the helpful comments. As can be seen from the detailed responses below, we intend to carefully consider all comments and will aim to adequately address them by carefully selected additional model simulations, analysis, and changes to the manuscript. Clearly one practical constraint that we need to sensibly account for is computational load and related efforts; our suggested ensemble extensions and advancements try to deal with this challenge in a best possible way. In the responses below, our comments are inserted with italicized, black text. The changes in the manuscript text are inserted in* green. *The original response from the referee is in* blue.

In their paper, Hohmann et al. studied the runoff sensitivity to spatial rainfall variability, namely they investigated the impact of station density and interpolation schemes on runoff simulations. I found the novelty of the study very limited. The effects of the density of the network of rain gauges and different interpolation methods on hydrological models have been analyzed many times in the past – as the authors mention themselves in the introduction to the paper. The research questions asked in the paper ("how many stations do we need to reliably model hydrological runoff under heavy rainfall events?" and "how large is the influence of interpolation scheme on runoff results under different station network densities?") are simply not new.

*Thank you for your comment. While we agree that the research questions are in general not new, we summarize here three main points of novelty we see and wanted to publish in this original "initial study approach". We also include some points on how we intend to advance the study design and improve the manuscript.*

1. *As far as we know, the combination of station density and interpolation scheme in the context of runoff at high resolutions (1 to 10 km scales) has not been studied in this way. In the manuscript we mention in the introduction (Lines 72 – 74) the studies of Gervais et al. (2014), Avila et al. (2015), and Herrera et al. (2018) which combine the impact of station density and interpolation scheme on rainfall data quality, but do not go the next step in the direction of integrating hydrology (i.e., sensitivity of runoff). So, we see new added value especially in the combination between the station density and interpolation scheme in the context of process based hydrological modelling at these high resolutions, with a particular focus to highly variable convective rainfall variability.* → *We will definitely have a closer look to the manuscript to strengthen this focus target even more. For example, we intend changes such as indicated in the following sentence:*

   L72 – 75 Since *previous studies have assessed the impact of the station density and interpolation on precipitation data quality such as mean and extreme rainfall values (Gervais et al., 2014; Avila et al., 2015; Herrera et al., 2018). We* go a next step and *focus on the impact of such precipitation uncertainty on hydrologic simulations, especially runoff peaks and the combination of station density and interpolation method.*

2. *Even though the method of studying sensitivities is obviously not new, we find that our results are. Especially given that we have a station (sub)networks and (sub)catchments design and combined rainfall event with hydrological modeling, something that was not possible in other regions than WegenerNet so far. We admit though that the design of the rainfall event ensemble, the subnetworks, and the interpolation benefits from further significant extension and improvement, which we now suggest below to implement during revision. The other*

*studies who analyzed station densities (e.g., Dong et al., 2005; Bárdossy and Das, 2008; Meselhe et al., 2009; Xu et al., 2013; Zeng et al., 2018; Huang et al., 2019) mention a threshold with no significant increase of model performance with a denser network. In our study we analyzed as one focus short-term convective events which do not show such an expected threshold especially for small catchments. Only the long duration stratiform rainfall events show such a threshold, with no improvement with more stations. The fact that the short convective events do not show this threshold behavior was, in our perception relative to the previous studies, one of the "unexpected" new results. → As an improvement of the study we intend to include a larger rainfall event ensemble, in line with reviewer suggestions, to further investigate and assess the robustness of such "unexpected" behavior. Specifically, we intend to use an ensemble of 20 to 30 short-duration events and 20 to 30 long-duration events. We will also strengthen the manuscript by discussing these events and results in more detail.*

3. *The highly dense precipitation station dataset, with 150 gauging station per 300 km² in its core region (complemented by eight operational network stations over 1000 km$^2$), is also a new opportunity to explore our study questions. Another quite dense network was used by Lopez et al. (2015), with 12 station per 1000 km²; and even though they added another 40 hypothetical stations in the Thur basin (~1700 km²), this study is still not as dense as ours. The other studies show even smaller station densities. → With an improved design in combination of a bigger rainfall event ensemble and a more and improved station subnetworks ensemble we can better capitalize on this novel network density for such a study. And we may point to this fact and the station data density in the manuscript in the introduction:*

L 65-67 The highly dense gridded station network WegenerNet (WEGN) (about 1 station every 2 km² over an area of 300 km$^2$) in the southeastern Alpine forelands of Austria allows us to study these questions related to the Raab catchment and its subcatchments.
L68-69 Because of the exceptionally dense data availability (Sect. 2.2) it is possible to analyze the influence of precipitation station densities on runoff in detail.

I found the results to be very limited following the decision to explore only two interpolation methods (TP and IDW), which seems to be motivated mainly by the fact that these are the interpolation methods that are built-into the WaSiM model that was used in this study while ignoring other common interpolation methods (e.g. Kriging).

*In WaSiM it is in principle feasible to include various external grids for meteorological input. Therefore, we confirm, Kriging was not excluded because it is not supported. We even tested Kriging, but did not see additional value. We more had seen the problem of how to decide which variogram to use, especially for a long-time frame of many years and a half-hourly resolution.*

*And for the purpose of exploring the key issue how individual-station rainfalls (point-scale time series) are spread by a certain interpolation method into the space, we have a new idea after receiving all three reviews: The IDW with exponent 2 will be kept as a baseline, and in contrast exponent 1 (quite more spread vs exponent 2) or exponent 3 (quite less spread vs exponent 2), provides the essential insights needed. Based on the evidence we have seen; we believe the key for the area rainwater flux received into the (sub)catchments is certainly how the spatial spreading plays out. Hence in the revision we tend to "simply" make our concept of systematic testing of interpolation influence really more clear. And we definitely will improve our interpolation setup per station subnetwork case, in particular that the overall catchment region around the WegenerNet core region (i.e., the general Raabtal region covered by the eight stations of operational ZAMG+AHYD stations) is kept at baseline*

*settings (also for interpolation). The subnetwork's densification is properly accompanied at each station density level by adequate interpolation settings (e.g., IDW2, IDW1 vs IDW2, IDW3 vs IDW2).*

*But again, having said the above, we will have a closer look as well to the use of Kriging. And we will find a reasonable way to use the interpolation method of ordinary Kriging as a possible alternative of IDW3 in our revised study setup. This will help to have a third interpolation method (Thiessen, IDW2, and Kriging), but still keep the computational time of the model feasible (currently around 4-5 days per simulation run per server at our computational servers). The computational time otherwise would "explode" with too many interpolation-method cases on top of significantly enlarged rainfall event and station subnetwork ensembles.*

*We are interested in the reviewer's opinion as to whether the systematic assessment of "spatial spread influence" of interpolations is not in his/her view also usefully covered already in the context of this study by the "IDW2 plus IDW1-vs-IDW2 and IDW3-vs-IDW2" approach. Especially now that we focus with this interpolation influence on the WegenerNet core region with its dense stations. Kriging and related issues may induce undue additional work and trials. We do not expect to additionally learn on the effect of how the increase of spatial spreading of point rainfalls impacts on runoff, beyond what we can learn from looking across the IDW1, IDW2, IDW3 cases at each subnetwork density level.*

Are six rainfall events enough for this type of study? Considering the authors discuss the effects of rainfall variability in space, I would expect many more events to be interpolated and simulated to demonstrate the robustness of the results.

*Thank for your comment. We also checked more events, but then decided to stick to these 6 ones, since they have been among the most extreme rainfall cases in our time frame.*

*As an improvement, we now intend to expand to an ensemble of events with 20 to 30 short-duration, heavy convective rainfall events and 20 to 30 long-duration heavy rainfall events. The selection of the events will still target the 10% heaviest rainfall events (i.e., above $90^{th}$ percentile in hourly intensity). To be able to include enough "really" heavy rainfall events the time period will be extended as needed from currently to 2012 out to 2018 or 2019 or so. With such change, which comes at significantly increased computational load and effort though, we have the opportunity to extract heavy/extreme events from a much larger pool of rainfall events. So clearly the robustness of the results will substantially increase compared to our "initial study approach" we followed so far.*

I cannot recommend the manuscript for publication in HESS in its current form. I would suggest extensive revisions of the study that include extending the number of extreme rainfall events in the analysis, exploring additional interpolation methods and, as the authors suggested in the abstract, conducting further investigation to study if a rainfall ensemble can be used to decompose the effects of rainfall records and interpolation uncertainties on modelled runoff – this might be qualified as a novel aspect that can contribute to the originality of the study, which is now missing.

*Thank you for this valuable list of suggestions how to improve the manuscript. In the following, we summarize the extensions and advancements we intend to implement in order to improve the study design and the manuscript along these suggestions.*

- *As one important point, already alluded to in the answers above, we will extend the number of extreme rainfall events significantly. We will use an ensemble of 20 to 30 short-duration*

*heavy rainfall events and 20 to 30 long-duration heavy rainfall events, that is 40 to 60 events in total. For more details see the notes above. This event-ensemble approach, compared to the handful of events of our "initial study approach", will help to statistically and in more detail analyze the effect of different station subnetwork densities. This clearly will give substantially more robustness to our results.*

- *As to a more systematic study of the co-influence of interpolation method choices, at each subnetwork density level, we want to test the "IDW2 plus IDW1-vs-IDW2 and IDW3-vs-IDW2" as a baseline. We, however, would keep the Thiessen as well and want to test also ordinary Kriging in our revised study setup (such ordinary Kriging assumes that the constant mean is unknown, which is reasonable in our case). This makes (at least) five interpolation-method options (at the limit of computational feasibility together with the other demands). We intend to decide what we really formulate into the revised manuscript based on the preliminary results. We are in particular interested to this end what the reviewer's opinion is as to whether the "IDW2 plus IDW1-vs-IDW2 and IDW3-vs-IDW2" alone wouldn't be sufficient to learn the essential effects of how the degree of spatial spreading of station rainfall data would impact the runoff results.*

- *We will significantly improve the station subnetwork density sampling, to strengthen the study setup overall. We still need to accept overall computational load limits. So, we will make the sequence of subnetwork cases denser and more systematic as follows: we still take the 5, and 5+3=8 operational ZAMG&AHYD station cases as the "background network" baseline, and otherwise only increment from one subnetwork to the next within a factor of 1.5 (rather than 2 or more), where the factor 1.45 was found helpful as a guide. This leads to a cascade of overall ten subnetworks with station number as follows: 5, 8, 12, 17, 25, 36, 56, 75, 109, 158 (with the seven number-cases 12, 17, 25, 36, 56, 75, 109 being the core of interim cases between just the operational network of 8 stations and the full 158 stations).*

- *In addition, regarding these ten subnetworks sizes, we intend to use two different subnetworks (spatially complementary, using different actual stations from the WEGN) each for the seven interim number-cases (i.e., from 12 to 109 stations) in order to also have a sensitivity crosscheck to actual spatial station distribution at a given total number of stations. Hence in total 17 subnetwork cases need to be analyzed within this design which together with the ensemble expansions in rainfall events and on interpolation choices stretches the computational load and efforts to the feasible limits. We agree that also this more balanced selection of subnetwork cases will substantially contribute to improved robustness of results.*

*With all these improvements, that are the best-possible tradeoff also regarding limits of computational load and efforts, we hope that the manuscript could potentially be recommended for publication since in this case we can more robustly and valuably help to better understand the sensitivity of high-flow runoff to highly variable rainfall records and to interpolation choices.*

*We are interested in the reviewer's opinion as to whether he/she considers that these are adequate improvements and so encourages us to pursue a revision along these lines.*

**Some specific comments:**

Line 150. What is the reason to calibrate the model with an objective function that includes both NSE and KGE? Why not simply using one of the two?

*The objective function NSE is more considering the peaks and the KGE more the whole water balance. Both of these aspects are important, especially if we want to analyze different types of rainfall runoff events (short-duration/long-duration). While calibrating the model, we also realized a different behavior of the objective function e.g. one parameter set lead to an improvement of the NSE, but a decrease of the KGE, or the other way around.*

Lines 156-157. The fact that the model is calibrated using IDW2 is not affecting the results when compared later to input using IDW3 and TP? I would expect the model to be calibrated to each input separately to reduce the errors emerging from the model parameterization (as your goal is to discuss the errors emerging from the model input).

*Thank you for the comment, we had the same thoughts, but comparing the model efficiency for the calibration period (Mai to September 2009) and validation period (Mai to September 2010) for all cases, the NSE value and KGE are almost the same with very little deviation to the calibration case (IDW2 158 Stations).  → We will make this clearer in the manuscript and add the table to the appendix.*

L 155 – 158: Because of the necessity of such manual recalibration, the model was calibrated with the IDW2 interpolation and 158 precipitation stations and not recalibrated with all different precipitation inputs and interpolation schemes. This setup is assumed to capture the spatial variation of precipitation in our study area. When comparing the NSE and KGE values for all cases, the deviation to the calibration run was found very small, with deviations for the calibration/ validation period exhibiting a maximum deviation of 0.02/ 0.07 in NSE and of 0.05/ 0.08 in KGE, respectively.

*Table 1: NSE and KGE efficiencies for the calibration period (Mai to September 2009) and validation period (Mai to September 2010) for all station numbers and interpolation method at gauging station Neumarkt/Raab. The model was calibrated with the IDW2 158 stations case at gauging station Neumarkt/Raab*

|  | Calibration Period Mai - Sep. 2009 | | Validation Period Mai - Sep. 2010 | |
|---|---|---|---|---|
|  | NSE | KGE | NSE | KGE |
| IDW2 158 Stations | 0.79 | 0.75 | 0.67 | 0.81 |
| TP 158 Stations | 0.81 | 0.76 | 0.65 | 0.81 |
| IDW3 158 Stations | 0.79 | 0.76 | 0.66 | 0.81 |
| IDW2 64 Stations | 0.79 | 0.74 | 0.66 | 0.81 |
| TP 64 Stations | 0.8 | 0.76 | 0.65 | 0.81 |
| IDW3 64 Stations | 0.8 | 0.75 | 0.66 | 0.81 |
| IDW2 32 Stations | 0.8 | 0.75 | 0.68 | 0.81 |
| TP 32 Stations | 0.81 | 0.77 | 0.66 | 0.81 |
| IDW3 32 Stations | 0.8 | 0.76 | 0.67 | 0.81 |
| IDW2 16 Stations | 0.8 | 0.72 | 0.69 | 0.81 |
| TP 16 Stations | 0.8 | 0.73 | 0.69 | 0.82 |
| IDW3 16 Stations | 0.8 | 0.73 | 0.69 | 0.81 |
| IDW2 8 Stations | 0.8 | 0.7 | 0.66 | 0.79 |
| TP 8 Stations | 0.79 | 0.75 | 0.63 | 0.77 |
| IDW3 8 Stations | 0.8 | 0.7 | 0.66 | 0.79 |
| IDW2 5 Stations | 0.8 | 0.76 | 0.64 | 0.76 |
| TP 5 Stations | 0.78 | 0.78 | 0.6 | 0.73 |
| IDW3 5 Stations | 0.8 | 0.77 | 0.62 | 0.75 |

Section 3.2.2. I suggest also comparing the rainfall events using some statistics. For example, how similar how the short rainfall events in term of their spatial variability? You can, for example, plot a graph comparing the spatial autocorrelation of the storms.

*Thank you for this suggestion. We intend to compute temporal and spatial variations using the standard deviation of rainfall amount (both normalized to the total rainfall amount) and spatial variation at the peak hour (normalized to the peak hourly rainfall amount). These numbers will help to compare short-duration versus long-duration events, but do not come into troubles with robust auto-correlations across events with varying duration.*

Section3.2.4. Why using only a single metric to analyze the runoff? Consider adding other metrics, pointing on the timing of the peak runoff, total runoff volume, etc.

*Thank you also for this suggestion. We had a short look to the timing of the runoff peak, but just saw little deviations for this specific event. But now, especially when adding many more events, we will analyze the timing of the peak runoff and include it in our results and discussion. We will also have a look to the total runoff volume and further investigate such different metrics to strengthen the robustness of the results.*

Lines 405-406. One option to overcome the model default is to compute the IDW2 outside the WaSiM model and a second option can be changing the code of WaSiM to compute the IDW the way you choose (if I recall the model as an open code policy).

*Yes, we also see this point and checked again the possibilities. Now we found an option to separate the areas for the interpolation, including a core area/zone that corresponds to the WEGN region and the area/zone around it, which is not covered be the dense network.*

*As a further improvement of the study setup, we intend to only change this WEGN area/zone in the precipitation input. Therefore, we will set the surrounding areas to one baseline setup of precipitation input (like the 8 Stations ZAMG&AHYD case with IDW2 interpolation) and then only change within the WEGN area/zone. We will create the precipitation input maps, which only show changes in this area, where we have the opportunity to well control the density of the setup. Hence, we can have a better focus on the area where we have additional information each time when the subnetwork is densified. In this way we also can more adequately adjust the maximum distance of IDW as needed, without inducing undue/unhelpful change in information for the surrounding (low density) stations and areas.*

---

## Author Comment (AC2) · 18 Nov 2020

**Response to Referee #2**

*We thank the reviewer for the helpful comments. As can be seen from the detailed responses below, we intend to carefully consider all comments and will aim to adequately address them by carefully selected additional model simulations, analysis, and changes to the manuscript. Clearly one practical constraint that we need to sensibly account for is computational load and related efforts; our suggested ensemble extensions and advancements try to deal with this challenge in a best possible way. In the responses below, our comments are inserted with italicized, black text. The changes in the manuscript text are inserted in* green. *The original response from the referee is in* blue.

This study tried to explore the effect of interpolation methods and station density on runoff simulation. However, this work has some severe problems and does not reach the standard of HESS in the current form. My comments are as follows.

**Major comments:**

1. It can hardly say that this study is novel or contributes much to our knowledge. The authors used two simple and widely used interpolation methods, TP and IDW. Results and conclusions are generally similar to previously published work.

*Thank you for your comment. Here we summarize three main points of novelty which we see and wanted to publish based on our original "initial study approach". We also include some points on how we intend to advance the study design and improve the manuscript.*

1. *As far as we know, the combination of station density and interpolation scheme in the context of runoff at high resolutions (1 to 10 km scales) has not been studied in this way. In the manuscript we mention in the introduction (Lines 72 – 74) the studies of Gervais et al. (2014), Avila et al. (2015), and Herrera et al. (2018) which combine the impact of station density and interpolation scheme on rainfall data quality, but do not go the next step in the direction of integrating hydrology (i.e., sensitivity of runoff). So, we see new added value especially in the combination between the station density and interpolation scheme in the context of process based hydrological modelling at these high resolutions, with a particular focus to highly variable convective rainfall variability. → We will definitely have a closer look to the manuscript to strengthen this focus target even more. For example, we intend changes such as indicated in the following sentence:*

   L72 – 75 Since *previous studies have assessed the impact of the station density and interpolation on precipitation data quality such as mean and extreme rainfall values (Gervais et al., 2014; Avila et al., 2015; Herrera et al., 2018). We* go a next step and *focus on the impact of such precipitation uncertainty on hydrologic simulations, especially runoff peaks and the combination of station density and interpolation method.*

2. *Even though the method of studying sensitivities is obviously not new, we find that our results are. Especially given that we have a station (sub)networks and (sub)catchments design and combine rainfall events with hydrological modeling, something that was not possible in other*

*regions than the WegenerNet so far. We admit though that the design of the rainfall event ensemble, the subnetworks, and the interpolation benefits from further significant extension and improvement, which we now suggest below to implement during revision. The other studies who analyzed station densities (e.g., Dong et al., 2005; Bárdossy and Das, 2008; Meselhe et al., 2009; Xu et al., 2013; Zeng et al., 2018; Huang et al., 2019) mention a threshold with no significant increase of model performance with a denser network. In our study we analyzed as one focus short-term convective events which do not show such an expected threshold especially for small catchments. Only the long duration stratiform rainfall events show such a threshold, with no improvement with more stations. The fact that the short convective events do not show this threshold behavior was, in our perception relative to the previous studies, one of the "unexpected" new results. → As an improvement of the study we intend to include a larger rainfall event ensemble, in line with reviewer suggestions, to further investigate and assess the robustness of such "unexpected" behavior. Specifically, we intend to use an ensemble of 20 to 30 short-duration events and 20 to 30 long-duration events. We will also strengthen the manuscript by discussing these events and results in more detail.*

3. *The highly dense precipitation station dataset, with 150 gauging station per 300 km² in its core region (complemented by eight operational network stations over 1000 km$^2$), is also a new opportunity to explore our study questions. Another quite dense network was used by Lopez et al. (2015), with 12 station per 1000 km²; and even though they added another 40 hypothetical stations in the Thur basin (~1700 km²), this study is still not as dense as ours. The other studies show even smaller station densities. → With an improved design in combination of a bigger rainfall event ensemble and a more and improved station subnetworks ensemble we can better capitalize on this novel network density for such a study. And we may point to this fact and the station data density in the manuscript in the introduction:*

L 65-67 The highly dense gridded station network WegenerNet (WEGN) (about 1 station every 2 km² over an area of 300 km$^2$) in the southeastern Alpine forelands of Austria allows us to study these questions related to the Raab catchment and its subcatchments.
L68-69 Because of the exceptionally dense data availability (Sect. 2.2) it is possible to analyze the influence of precipitation station densities on runoff in detail.

Besides, as a study focusing on the effect of station density and interpolation methods on hydrological modeling, the design is too simple. It is better and not difficult to (1) design station networks with random numbers and locations of stations instead of using pre-defined networks (Table 3), and (2) conduct a sensitivity test of the weighting power instead of using 2 and 3. A negative example caused by (1) is that the authors state "32 stations would be enough . . .", but this resulted from a large jump from 16 to 32 to 64 stations.

*Thank you very much for the suggestion. We first explain our thoughts behind these points and then also give our suggestions how to improve.*

*To point 1) We selected the station densities in this way: Our base setup is the full network with 158 stations, we assume to catch the precipitation events in the best possible way. Then, step by step we reduced the number of stations by removing them randomly, while ensuring an uniform spatial distribution by considering the area-of-influence of each station (see also supplement of O and Foelsche 2019 https://doi.org/10.5194/hess-23-2863-2019-supplement -> Rain-gauge sub-networks for more details about the method). Again, we assume to catch the precipitation with these well*

*distributed setups as good as possible. We end up with the 8 and 5 stations case, which is the operational stations network (of the meteorological and hydrological services of Austria, ZAMG and AHYD), without any stations from the research network. With this, we tried to get a reasonable setup of stations to cover the area as good as possible. The idea of doubling the station number was based on our expectation as well as based on literature, that the biggest uncertainties/sensitivities are observed for fewer stations, where the "jumps" are smaller.*

*However, we will significantly improve the station subnetwork density sampling, to strengthen the study setup overall. We still need to accept overall computational load limits. → So, we will make the sequence of subnetwork cases denser and more systematic as follows: we still take the 5, and 5+3=8 operational ZAMG&AHYD station cases as the "background network" baseline, and otherwise only increment from one subnetwork to the next within a factor of 1.5 (rather than 2 or more), where the factor 1.45 was found helpful as a guide. This leads to a cascade of overall ten subnetworks with station number as follows: 5, 8, 12, 17, 25, 36, 56, 75, 109, 158 (with the seven number-cases 12, 17, 25, 36, 56, 75, 109 being the core of interim cases between just the operational network of 8 stations and the full 158 stations).*

*We also see the potential utility of a selection of many more station networks. But the option with random picking, and hundreds of runs is really not feasible for our setup, because of computational time with the process-based modelling approach. → To further investigate the uncertainty also related to pre-defined subnetworks of given number, we intend to use two different subnetworks (spatially complementary, using different actual stations from the WEGN) each for the seven interim number-cases (i.e., from 12 to 109 stations). Thus, we have a sensitivity crosscheck to actual spatial station distribution at a given total number of stations. Hence in total 17 subnetwork cases need to be analyzed within this design which together with the ensemble expansions in rainfall events and on interpolation choices stretches the computational load and efforts to the feasible limits. We agree that also this more balanced selection of subnetwork cases will substantially contribute to improved robustness of results.*

*To point 2) Thank you for this suggestion, which we also will have to deal with a due eye to feasibility. For the purpose of exploring the key issue how individual-station rainfalls (point-scale time series) are spread by a certain interpolation method into the space, we would like to study this now in the following way: The IDW with exponent 2 will be kept as a baseline, and in contrast exponent 1 (quite more spread vs exponent 2) or exponent 3 (quite less spread vs exponent 2). Based on the evidence we have seen; we believe the key for the area rainwater flux received into the (sub)catchments is certainly how the spatial spreading plays out. Hence in the revision we tend to "simply" make our concept of systematic testing of interpolation influence really clearer. And we additionally will improve our interpolation setup per station subnetwork case, in particular that the overall catchment region around the WegenerNet core region (i.e., the general Raabtal region covered by the eight stations of operational ZAMG+AHYD stations) is kept at baseline settings (also for interpolation). The subnetwork's densification is properly accompanied at each station density level by adequate interpolation settings (e.g., IDW2, IDW1 vs IDW2, IDW3 vs IDW2).*

*We are interested in the reviewer's opinion as to whether the systematic assessment of "spatial spread influence" of interpolations is not in his/her view usefully covered already in the context of this study by the "IDW2 plus IDW1-vs-IDW2 and IDW3-vs-IDW2" approach. Especially now that we focus with this interpolation influence on the WegenerNet core region with its dense stations. Instead of including Thiessen Polygons and/or additional interpolation schemes like Kriging. Since we do not expect to additionally learn on the effect of how the increase of spatial spreading of point rainfalls*

*impacts on runoff, beyond what we can learn from looking across the IDW1, IDW2, IDW3 cases at each subnetwork density level.*

2. A large part of the study area is not covered by 158 rain gauges. Both IDW and TP cannot achieve reasonable estimates outside the WEGN network. As a result, only rainfall in the middle reach is largely affected by different density and interpolation methods, while rainfall in the upper and lower reach could always contain much larger errors. Meanwhile, the hydrological model is built over the whole domain, and runoff at catchment and sub-catchment outlets is compared. The effect of biased estimates in upper and lower reach on the runoff simulation could be very large.

*Yes, that is true. To overcome this uncertainty, we choose only subcatchments, which were mostly covered by WEGN. Since the subcatchments are in focus, we should make this clearer. And it is true, that we still should mention this uncertainty of the missing coverage for gauging station Neumarkt/Raab and Feldbach/Raab in the manuscript. We intend to do so.*

*Another aspect is that the setup with the 5 or 8 stations (around 5 or 8 stations per 1000km²) with well distributed gauges over the total catchment is already more than other studies have to choose from. E.g. Xu et al. (2013) studied the Xiangjiang River (94 660 km²) with around 2 rain gauges per 1000 km² (threshold with 1 station per 1000 km²). Dong et al. (2005) studied the Qingjiang river (12 209 km2) with also with around 2 rain gauges per 1000 km² (threshold around 0.4 gauges per 1000 km²).*

*To overcome the point that only the middle reach is affected, we intend to only change this WEGN area/zone in the precipitation input. Therefore, we will set the surrounding areas to one baseline setup of precipitation input (like the 8 Stations ZAMG&AHYD case with IDW2 interpolation) and then only change within the WEGN area/zone. We will create the precipitation input maps, which only show changes in this area, where we have the opportunity to well control the density of the setup. Hence, we can have a better focus on the area where we have additional information each time when the subnetwork is densified. In this way we also can more adequately adjust the maximum distance of IDW as needed, without inducing undue/unhelpful change in information for the surrounding (low density) stations and areas.*

3. Please perform the analysis based on a larger collection of precipitation events. Currently, only three small-scale short-duration and three large-scale long-duration events are selected. Although the six events may be representative, a comprehensive view in a long historical period is useful and necessary to demonstrate the all-aspect effect of interpolation and station density. The current results and analyses are all based on those limited events, making the results more "casual" than "causal". For example, the interpolation based on 5 stations could be largely affected by the location of storm centers, and thus the results based on 100 events could be different with results based on 6 events.

*Thank for your comment. Yes, it is true, they might be affected. We also checked more events, but then decided to stick to these 6 ones, since they have been among the most extreme rainfall cases in our time frame.*

*As an improvement, we now intend to expand to an ensemble of events with 20 to 30 short-duration, heavy convective rainfall events and 20 to 30 long-duration heavy rainfall events. The selection of the events will still target the 10% heaviest rainfall events (i.e., above $90^{th}$ percentile in hourly intensity). To be able to include enough "really" heavy rainfall events the time period will be extended as needed from currently 2012 out to 2018 or 2019 or so. With such change, which comes at significantly*

*increased computational load and effort though, we have the opportunity to extract heavy/extreme events from a much larger pool of rainfall events. So clearly the robustness of the results will substantially increase compared to our "initial study approach" we followed so far.*

*With this approach, we hope to be more representative for the study area. And we also hope to overcome the problem with the location of single storm cells a bit more.*

4. Figure 5: The IDW precipitation map based on 158 stations looks quite unrealistic. An actual precipitation event should be spatially continuous like that in Figure 4. However, the bull eye effect in Figure 5 is too obvious. The authors stated that all stations within a 50 km searching radius would be used in IDW. 50 km is quite large based on the measuring scale in Figure 1. This makes the bull eye effect even weirder.

*We will double check this point. But to overcome the problem with just one search radius, we will (as already mentioned in point 2.) only change within the WEGN area the different interpolation schemes and network densities. Now, we will also have the option to change the search radius for every network density respectively. We also hope to overcome the "bull eye effect" with this new setup a bit more, to the degree that area rainfall flux into subcatchments appears clearly plausible.*

5. The authors strengthened "spatial rainfall variability" in the title. But the spatial variability is not explicitly analyzed in this study. The title does not reflect what this study actually did, i.e., station density and interpolation methods. Besides, the effect of station density and interpolation methods cannot be simply represented using "spatial variability".

*Thank you for pointing this out. We will consider changing the title.*

**Specific comments:**

6. Please adjust the font of units which is often different from that of texts.

*Thank you for the hint, it might be because the units are introduced in LaTeX with \unit{XX} e.g. \unit{km}, like the Copernicus LaTeX package said.*

*"%% Copernicus Publications Manuscript Preparation Template for LaTeX Submissions*

*%%% PHYSICAL UNITS: Please use \unit{} and apply the exponential notation"*

*But we will try to solve this problem.*

7. Line 25: It is better to state the "measurement uncertainties of rain gauges" because other approaches of rainfall measuring are not mentioned here.

*We will change the sentence to:*

*(Line 24-26)* Beside the measurement uncertainties of rain gauges, considerable uncertainty can arise when point-level measurements are spatially interpolated for final gridded products (Goodrich et al., 1995; Mcmillan et al., 2012; Huang et al., 2019; O and Foelsche, 2019).

**8. Line 32-34: The pros and cons of radars and satellite sensors are similar in many cases and should be stated together.**

*We will change the sentences to:*

*(Line 31-34)* On the other side, indirectly estimate precipitation like radar systems and satellites show a higher spatial resolution of the precipitation cells, but do not give specific precipitation amounts (e.g., Sun et al., 2000; Tetzlaff and Uhlenbrook, 2005). They indirectly estimate precipitation and therefore their data are subject to errors and uncertainties (e.g., Sun et al., 2000; Tetzlaff and Uhlenbrook, 2005; Tian and Peters-Lidard, 2010; Kirstetter et al., 2012; O et al., 2017; Lasser et al., 2019).

**9. Figure 1: It will be helpful to add latitude and longitude.**

*Thanks, yes, we will add latitude and longitude to both maps.*

**10. Line 124: Please complete the reference for WaSiM.**

*Sorry for that. We will correct the sentence accordingly:*

We used the hydrological model WaSiM, developed by Schulla et al. (1997), at the ETH Zurich in Switzerland for climate change studies in Alpine catchments.

**11. Line 150: Although I understand what you mean by 50% NSE and 50% KGE, please rephrase to be more formal.**

*Thank you. We attend to rewrite the sentence e.g. like this:*

The model performance was assessed with 50 % the Nash-Sutcliffe efficiency (NSE) (Nash and Sutcliffe, 1970) and 50 % the Kling-Gupta efficiency (KGE) (Gupta et al., 2009) both weighted half of the total model efficiency.

**12. Line 155-162: Given the authors state that manual recalibration is necessary, please add some descriptions on the benefits of manual recalibration. For example, what's the KGE and NSE before manual recalibration? Besides, the NSE decreased from calibration to validation periods but KGE increased. Please add some explanations.**

*The efficiency measures NSE and KGE were pretty good with the SCE-UA, since they were the goal of the algorithm. But with this parameter set, the runoff components were physically unrealistic, e.g., with no baseflow at all, which is not realistic for our catchment in the Alpine foreland. Therefore, we did a manual recalibration with a focus on the runoff components.*

*We will make this clearer in the revised manuscript and add some more lines about our calibration strategy.*

**13. Line 167: Strictly speaking, you used two, not three methods. Two different parameters do not make IDW two different methods.**

*Yes, this will be rephrased, depending on the new setup of interpolation methods.*

14. Figure 4: Please use the shapefile of the catchment to replace the black box, which can help identify whether storm centers are located within or outside the river basin.

*Thank you for this hint. We will change the figure, including the catchment appearance.*

---

## Author Comment (AC3) · 18 Nov 2020

**Response to Referee #3**

*We thank the reviewer for the helpful comments. As can be seen from the detailed responses below, we intend to carefully consider all comments and will aim to adequately address them by carefully selected additional model simulations, analysis, and changes to the manuscript. Clearly one practical constraint that we need to sensibly account for is computational load and related efforts; our suggested ensemble extensions and advancements try to deal with this challenge in a best possible way. In the responses below, our comments are inserted with italicized, black text. The changes in the manuscript text are inserted in green. The original response from the referee is in blue.*

**General comments and recommendation**

The manuscript by Hohmann et al. presents an analysis of the effect of different meteorological network densities, as well as different interpolation schemes on the simulated runoff for a meso-scale catchment in southeastern Austria. While indeed many questions about the optimal meteorological network density, about the adequacy and representativeness of stations' distribution and about the suitability of different interpolation schemes for hydrological modelling have not yet been fully answered, I do not see right now in this study an adequate and robust assessment, offering an added value to what we already know. The setup and design of the experiment is not thorough enough to support reliable statements with evidence.

*Thank you very much for seeing the importance of an "optimal meteorological network density" in the context of hydrological modelling. We would like further strengthen the setup and study design to be able to support reliable statements with more evidence. Here we will just name our ideas and further describe these points at the specific comments below.*

- *Include an ensemble of events with 20 to 30 short-duration and 20 to 30 long-duration events*
- *Strengthen the stations networks, with more setups: 5, 8, 12, 17, 25, 36, 56, 75, 109 and 158 stations*
- *Include a second setup of station subnetworks of interim sizes (12 to 109), to further investigate the uncertainty of pre-defined networks with a given station number*
- *Redefine interpolation methods, and make their sensitivity investigation more systematic*

The manuscript has per se a clear structure, the methods are generally described in a comprehensible way or supported by relevant sources, however, some explanations could be clearer and more concise (e.g. the calibration procedure). Even though the discussion provides some good points, generally there are quite a few redundant paragraphs, while more interesting and critical points are not examined closely enough. The manuscript generally features high-quality and interesting figures; some of the tables and their captions should be reorganized in a more meaningful and efficient way. I would also suggest a native speaker to read it and correct it, some sentences definitely need to be rephrased.

*Thank you for mentioning these points. We will revise the calibration chapter in our manuscript to further explain our calibration procedure. For more specific details see the point below under "Calibration of the hydrological model". The discussion part will be adjusted addressing the new study setup, including more critical discussion and avoiding redundant paragraphs. Also, the tables and captions will get a close look. And for the revised manuscript we will also ask a native speaker to read*

*through it. Thank you also for mentioning some sentences to be rephrased (see points below in the "Discussion" part).*

Because of these considerations, I think the manuscript requires and extension of the experimental design, a more critical discussion and further work, before it can be possibly recommended for publication.

*As you can see above in our first answer, we intend to improve our experimental design in several major points. We will also give more attention to the discussion part to further investigate the critical points. With all these improvements we hope that the manuscript could be recommended and we can help to more robustly answer open questions about the optimal precipitation network density and the suitability of different interpolation methods for hydrological runoff simulations.*

Please find my specific and technical comments here following.

**Specific comments**

- Experimental design:

  • One important drawback of your setup is the fact that despite the stations' density is high, it is not covering your whole study area, but only its central part.

    *As an explanation, the WEGN was buildup 2007 for meteorological and climate change setups, which led to have the near-rectangular domain, instead of covering a catchment domain.*

    *That the dense station network it is not covering the whole study area it true, but mostly effecting the total catchment, so the gauging stations Feldbach/Raab and Neumerkt/Raab. To overcome this uncertainty, we choose only subcatchments which were mostly covered by WEGN stations. Since the small-scale subcatchments are in focus, we agree that we need to make this clearer in the manuscript. And additionally, we will also mention this uncertainty of the missing coverage for gauging station Neumarkt/Raab and Feldbach/Raab in the manuscript.*

    *Beside these changes in the manuscript and to further overcome the uncertainty of the dense station network, compared to the surrounding stations, we intend to only change the WEGN area/zone in the precipitation input. Therefore, we intend to set the surrounding areas to one baseline setup of precipitation input (like the 8 Stations ZAMG&AHYD case with IDW2 interpolation) and then only change within the WEGN area/zone. We will create the precipitation input maps, which only show changes in this area, where we have the opportunity to well control the density of the setup. Hence, we can have a better focus on the area where we have additional information each time when the subnetwork is densified.*

  • You have "fix" a priori chosen subnetwork configurations, but for a fair evaluation of the effect of stations' subnetwork density you should try different (random?) configurations for the same number stations. Tentatively, you could also consider smaller jumps between one configuration and the next one.

    *We selected the station densities in this way: Our base setup is the full network with 158 stations, we assume to catch the precipitation events in the best possible way. Then, step*

*by step we reduced the number of stations by removing them randomly, while ensuring a uniform spatial distribution by considering the area-of-influence of each station (see also supplement of O and Foelsche 2019 https://doi.org/10.5194/hess-23-2863-2019-supplement -> Rain-gauge sub-networks for more details about the method). Again, we assume to catch the precipitation with these well distributed setups as good as possible. We end up with the 8 and 5 stations case, which is the operational stations network (of the meteorological and hydrological services of Austria, ZAMG and AHYD), without any stations from the research network. With this, we tried to get a reasonable setup of stations to cover the area as good as possible. The idea of doubling the station number was based on our expectation as well as based on literature, that the biggest uncertainties/sensitivities are observed for fewer stations, where the "jumps" are smaller.*

*However, we will significantly improve the station subnetwork density sampling, to strengthen the study setup overall. We still need to accept overall computational load limits. → So, we will make the sequence of subnetwork cases denser and more systematic as follows: we still take the 5, and 5+3=8 operational ZAMG&AHYD station cases as the "background network" baseline, and otherwise only increment from one subnetwork to the next within a factor of 1.5 (rather than 2 or more), where the factor 1.45 was found helpful as a guide. This leads to a cascade of overall ten subnetworks with station number as follows: 5, 8, 12, 17, 25, 36, 56, 75, 109, 158 (with the seven number-cases 12, 17, 25, 36, 56, 75, 109 being the core of interim cases between just the operational network of 8 stations and the full 158 stations).*

*We also see the potential utility of a selection of many more station networks. But the option with random picking, and hundreds of runs is really not feasible for our setup, because of computational time with the process-based modelling approach. → To further investigate the uncertainty also related to pre-defined subnetworks of given number, we intend to use two different subnetworks (spatially complementary, using different actual stations from the WEGN) for each of the seven interim number-cases (i.e., from 12 to 109 stations). Thus, we have a sensitivity crosscheck to actual spatial station distribution at a given total number of stations. Hence in total 17 subnetwork cases need to be analyzed within this design, which together with the ensemble expansions in rainfall events and on interpolation choices (see bloew), stretches the computational load and efforts to the feasible limits. We agree that also this more balanced selection of subnetwork cases will substantially contribute to improved robustness of results.*

- You chose 2 interpolation methods (I wouldn't refer to three methods, as you simply changed a parameter of the second method), I think it would be more appropriate to use more interpolation methods, such as kriging, etc.. for a more robust evaluation of the effect given by the interpolation method, and also definitely to use a smaller searching radius (second spurious peaks in your hydrological simulations are not surprising, given how you interpolate precipitation).

*Thank you for your comment. We tested Kriging, but did not see additional value. We more had seen the problem of how to decide which variogram to use, especially for a long-time frame of many years and a half-hourly resolution. But again, we will have a closer look as well to the use of Kriging. And we will find a reasonable way to use the interpolation method of ordinary Kriging. But we still have to deal with the computational time,*

*otherwise it would "explode" with too many interpolation-method cases on top of significantly enlarged rainfall event and station subnetwork ensembles.*

*Also for the purpose of exploring the key issue how individual-station rainfalls (point-scale time series) are spread by a certain interpolation method into the space, we have a new idea after receiving all three reviews: The IDW with exponent 2 will be kept as a baseline, and in contrast exponent 1 (quite more spread vs exponent 2) or exponent 3 (quite less spread vs exponent 2), provides the essential insights needed. Based on the evidence we have seen; we believe the key for the area rainwater flux received into the (sub)catchments is certainly how the spatial spreading plays out. Hence in the revision we tend to "simply" make our concept of systematic testing of interpolation influence really clearer. And we definitely will improve our interpolation setup per station subnetwork case, in particular that the overall catchment region around the WegenerNet core region (i.e., the general Raab catchment covered by the eight stations of operational ZAMG+AHYD stations) is kept at baseline settings (also for interpolation). The subnetwork's densification is properly accompanied at each station density level by adequate interpolation settings (e.g., IDW2, IDW1 vs IDW2, IDW3 vs IDW2).*

*We are interested in the reviewer's opinion as to whether the systematic assessment of "spatial spread influence" of interpolations is not in his/her view also usefully covered already in the context of this study by the "IDW2 plus IDW1-vs-IDW2 and IDW3-vs-IDW2" approach. Especially now that we focus with this interpolation influence on the WegenerNet core region with its dense stations. Kriging and related issues may induce undue additional work and trials. We do not expect to additionally learn on the effect of how the increase of spatial spreading of point rainfalls impacts on runoff, beyond what we can learn from looking across the IDW1, IDW2, IDW3 cases at each subnetwork density level.*

*To be able to further adjust the search radius of the IDW interpolation we will split up the interpolation maps in the core area/zone of the WEGN and the surrounding area, which is not covered be the dense network. And then the idea is to only change this core WEGN area/zone in the precipitation input. The surrounding areas will be set to a baseline setup of precipitation input (like the 8 Stations ZAMG&AHYD case with IDW2 interpolation), which will not be changed. Therefore, will create precipitation input maps, which only show changes in the WEGN area/zone. With this approach we can adjust the search radius of IDW individually for each density network, without losing necessary information of surrounding precipitation stations.*

The number and sample of events you are analyzing is simply too limited to allow you to make any meaningful assessment, now your analysis might show only very specific and localized effects, and might not hold for a larger ensemble of events. Is there any way to increase the number of events you analyze? You say you selected first heavy precipitation events among the top 10% heaviest rainfall days during summer, and out of these you selected through visual inspection your 6 events. I would suggest you rather select the top 100 events for 1d and 3d accumulation periods, for instance, and further analyze these?

*Thank for this comment. We also checked more events, but then decided to stick to these 6 ones, since they have been among the most extreme rainfall cases in our time frame.*

*We agree with the reviewer that a larger ensemble of events will strengthen the robustness of our analysis. So, we now intend to expand to an ensemble of events with 20 to 30 short-duration, heavy convective rainfall events and 20 to 30 long-duration heavy rainfall events. The selection of the events will still target the 10% heaviest rainfall events (i.e., above 90th percentile in hourly intensity). To be able to include enough "really" heavy rainfall events the time period will be extended as needed from currently to 2012 out to 2018 or 2019. With such change, which comes at significantly increased computational load and effort though, we have the opportunity to extract heavy/extreme events from a much larger pool of rainfall events. So clearly the robustness of the results will substantially increase compared to our "initial study approach" we followed so far.*

- (You are only using discharge gauging stations on the main river trunk, but you analyze also the contributing subcatchments. Sure, this is fine for looking at the effect of the different precipitation inputs, but not enough to disentangle the effects possibly stemming from the parameters of the hydrological model. You implicitly assume the hydrological model is working equally well on much smaller subcatchments, with the same parameters. As the model is process oriented I am fine with this assumption, but you might want to spend a few words on this?)

*Thank you for pointing this out. Yes, unfortunately we do not have measured runoff data for the subcatchments. That is one reason why we decided to use a process-oriented model. We will also improve the description in the manuscript to this end.*

- **Data**:

- You have a 10 year period to choose from resp. analyze, is there some way to extend it?

*From the WEGN we have data from 2007 until now. We chose the time frame of 2009 to 2012, because of known especially extreme events in this period. We will check the extreme events in our time frame, and the time period will be extended as needed from currently to 2012 out to 2018 or 2019 or so. Another aspect we have to care about, is the computational time, so we tried to keep each model run as short as necessary.*

- You only report the return period for discharge, but actually it would be interesting – and relevant? - to know the return period associated with your rainfall events too.

*Thank you, we will look into this issue again, too. We consider that we only have data from 2007 onward within the WEGN. The runoff timeseries of station Neumarkt/Raab is quite longer, with data starting from 1991.*

- **Model setup:**

- P7-L128-130: Lumped doesn't necessarily mean that a model is not process-oriented. I guess you mean conceptual? (that are often lumped, but not always)

*Yes, exactly that is what we meant. We will change it to:*

Line 128-130: We focused on a process-oriented model to keep the model uncertainty small, compared to  conceptual models, which are often used for similar precipitation runoff studies (e.g., Dong et al., 2005; Zeng et al., 2018; Huang et al., 2019).

- **Calibration of the hydrological model**

    - Please describe better and provide more details on your calibration procedure.

    *Yes, we will provide such a paragraph with more details, and maybe include a section in the appendix to have the space to further describe the calibration without unduly lengthening the manuscript.*

    - Why do you calibrate the model only basing on one summer, and only for one interpolation method?

    *We calibrated the model just for one interpolation method, but we also checked the efficiencies for the other ones. In Table 1 you find the NSE and KGE efficiencies for the other interpolation methods and also different station densities.*

*Table 1: NSE and KGE efficiencies for the calibration period (Mai to September 2009) and validation period (Mai to September 2010) for all station numbers and interpolation method at gauging station Neumarkt/Raab. The model was calibrated with the IDW2 158 stations case at gauging station Neumarkt/Raab.*

|  | Calibration Period Mai - Sep. 2009 | | Validation Period Mai - Sep. 2010 | |
|---|---|---|---|---|
|  | NSE | KGE | NSE | KGE |
| IDW2 158 Stations | 0.79 | 0.75 | 0.67 | 0.81 |
| TP 158 Stations | 0.81 | 0.76 | 0.65 | 0.81 |
| IDW3 158 Stations | 0.79 | 0.76 | 0.66 | 0.81 |
| IDW2 64 Stations | 0.79 | 0.74 | 0.66 | 0.81 |
| TP 64 Stations | 0.8 | 0.76 | 0.65 | 0.81 |
| IDW3 64 Stations | 0.8 | 0.75 | 0.66 | 0.81 |
| IDW2 32 Stations | 0.8 | 0.75 | 0.68 | 0.81 |
| TP 32 Stations | 0.81 | 0.77 | 0.66 | 0.81 |
| IDW3 32 Stations | 0.8 | 0.76 | 0.67 | 0.81 |
| IDW2 16 Stations | 0.8 | 0.72 | 0.69 | 0.81 |
| TP 16 Stations | 0.8 | 0.73 | 0.69 | 0.82 |
| IDW3 16 Stations | 0.8 | 0.73 | 0.69 | 0.81 |
| IDW2 8 Stations | 0.8 | 0.7 | 0.66 | 0.79 |
| TP 8 Stations | 0.79 | 0.75 | 0.63 | 0.77 |
| IDW3 8 Stations | 0.8 | 0.7 | 0.66 | 0.79 |
| IDW2 5 Stations | 0.8 | 0.76 | 0.64 | 0.76 |
| TP 5 Stations | 0.78 | 0.78 | 0.6 | 0.73 |
| IDW3 5 Stations | 0.8 | 0.77 | 0.62 | 0.75 |

*We calibrated the model for the summer 2009, since in 2009 many extreme events occurred which are in focus of our study. Beside that we also wanted to keep the computational time in a reasonable way, to be able to do the first calibration steps with the SCE-UA. And as we mentioned before the high-resolution process-oriented model setup need quite a lot computational work.*

- Why do you use "only" NSE and KGE to define your objective function, instead of including further goodness-of-fit measures and criteria, perhaps more specifically tailored for floods, their volume and/or their timing?

*The objective function NSE is more considering the peaks and the KGE more the whole water balance. Both aspects are important, especially if we want to analyze different types of rainfall runoff events (convective/advective). We wanted to focus one these two aspects, but we can of cause also have a look to further goodness-of-fit measures.*

- **Runoff analysis approach:**
  - I would recommend you expand your analysis approach by including also the peak timing, as this also is an important aspect in your modelling exercise, e.g. considering the effect of superposition of peaks simulated in the subcatchments.

*Thank you also for this suggestion. For some events we even checked the peak timing and saw no huge difference, therefore we did not include it. But now, especially when adding many more events, we will analyze the timing of the peak runoff and include it in our results and discussion. Thereby we will have a second source to analyze the runoff behavior of the (sub)catchments.*

- **Results**:
  - P16-L271: I wouldn't say that, e.g. Kornbach is rather systematically overestimated.

*Yes we agree, we may delete the sentence, but we will see how the analysis results of the ensemble of events come up. This will help to further investigate the effect of overestimation or underestimation of specific sub-catchments.*

Line 270 to 271:

- P16-L275-277: this is not true, see for example the short-2 event or the long-3 event. –

*We wanted to say that, if we simulate the runoff with 5 Stations under the TP interpolation scheme and then with 8 Stations and the TP interpolation scheme we will get the same runoff. So, the same runoff deviation compared between the two runs. As an example: For the short-2 event the peak flow deviation will be in the Grazbach -11 for the 5-Stations case and -11 for the 8-Stations case. For the short-3 e.g. at Auersbach the peak flow deviation is 32 for the 5-Stations and 8-Stations case.*

*We will change the sentence to make this more clear and easier readable, so that no misunderstanding should occur.*

*Line 275-277:* The northern catchments Auersbach, Kornbach and Grazbach do not show differences if we simulate with 5- or 8-Stations subnetworks under the TP interpolation scheme, because of their location in relation to these station locations. *(Will be changed)*

**Discussion**:

- P 20: it is confusing you mentioning first Lopez et al.2015 report no increase in performance after including 24 gauges per 1000 km2, and later on saying they actually used 12 stations per 1000km2. They used gridded products to be able to increase the number of stations with "hypothetical" gauging stations. I think you should either specify this, or not go so much into detail.

*Yes, while working with the other reviewer comments we also recognized this confusion. We will make sure to clarify this point and adjust the manuscript.*

- P20-L360-363: Rephrase please this last paragraph.

*Yes, we will rephrase these sentences in the revised manuscript.*

- P21-L380-381: Isn't this possibly case specific? Refer also to Lobligeois et al. 2014: In all regions, natural variability allows for contradictory examples to be found, showing that analyzing a large number of events over varied catchments is warranted.

*We hope to be able to further investigate to this point, when we analyze the ensemble of events and then adjust the sentence in the manuscript.*

- I think you should expand more on the representativeness of stations' location.

*Good hint, we will add a paragraph in the discussion part. Besides the option of just one best distributed station network we will include a second setup with also a very good distribution of stations (see in the answers above). This will help to further analyze the stations location. This analysis will also be carefully discussed in the manuscript.*

**Technical corrections**

- P6-L124: developed..by who et al.?

*Sorry for this. We will correct the sentence:*

We used the hydrological model WaSiM, developed by Schulla et al. (1997), at the ETH Zurich in Switzerland for climate change studies in Alpine catchments.

- P7-L128: BAFU is FOEN in english

*Thank you, we will change the part to:*

[…] up to operational use (e.g. at  FOEN Switzerland).

- Figure 2: shouldn't you remove the box with snow accumulation/melt, as far as I understood you are not using it?

*The model is running with the snow module, to be able to simulate continually over time. But the focus time are only the summer month. So, it would be an option to exclude it, because it is not in focus and will not be calibrated.*

- Figure 4: You should be consistent, and either use the min-max range in both cases, or the 5th and 95th percentile range. And why don't you also show the same kind of information for the 3rd source of precipitation data?

*We intend to update the figure using min-max range for all three datasets.*

- Figure 6: Why don't you show observations in the last column?(i.e. for the Neumarkt/Raab gauging station)

*We wanted to keep the comparability between the different catchments. Another line in the Neumarkt/Raab figure was more confusing than supporting, from our experience. But we will double check this and may change this.*

- P17-L301 ..in these subcatchment should be in *this* subatchment

*Yes, we will correct the sentence accordingly:*

*Line 300-301:* Only the 64-Stations subnetwork under the IDW3 interpolation scheme result in a −25 % peak flow deviation in  this subcatchment.

- P21-L373: ..more event-depended should be event-*dependent*

*Yes, we will correct the sentence accordingly:*

*Line 373:* The runoff for the short-duration events is much more event- dependent